# LogiStory: A Logic-Aware Framework for Multi-Image Story Visualization

**Chutian Meng, Fan Ma, Chi Zhang, Jiaxu Miao, Yi Yang & Yueting Zhuang** [*]
College of Computer Science and Technology
Zhejiang University
Hangzhou 310027, China
`{12321215, mafan, 12321216, jiaxumiao, yangyics, yzhuang}@zju.edu.cn`

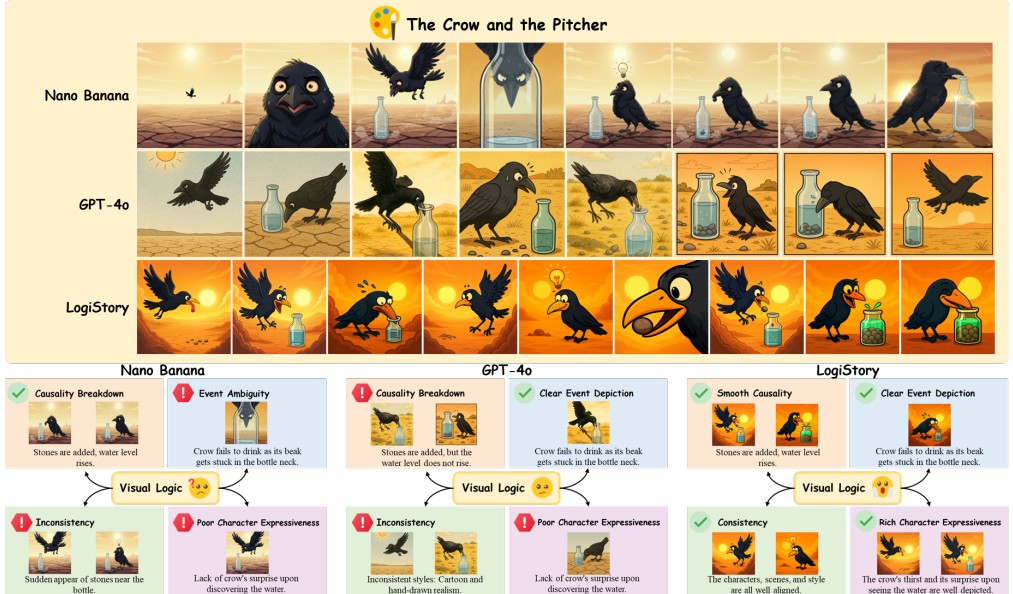

Figure 1: Comparison of the state-of-the-art multimodal models alongside our proposed approach LogiStory in the visualization of the simple story "The Crow and the Pitcher." The results highlight the challenges of visual reasoning in the process of visual sequence generation, while demonstrating the effectiveness of LogiStory.

## ABSTRACT

Generating coherent and communicative visual sequences, such as image sequences and videos, remains a significant challenge for current multimodal systems. Despite advances in visual quality and the integration of world knowledge, existing models still struggle to maintain logical flow, often resulting in disjointed actions, fragmented narratives, and unclear storylines. We attribute these issues to the lack of attention to **visual logic**, a critical yet underexplored dimension of visual sequence generation that we define as the perceptual and causal coherence among characters, actions, and scenes over time. To bridge this gap, we propose a logic-aware multi-image story visualization framework, **LogiStory**. The framework is built around the central innovation of explicitly modeling visual logic in story visualization. To realize this idea, we design a multi-agent system that grounds roles, extracts causal chains, and verifies story-level consistency, transforming narrative coherence from an implicit byproduct of image generation into an explicit modeling objective. This design effectively bridges structured story planning with visual generation, enhancing both narrative clarity and visual quality in story visualization. Furthermore, to evaluate the generation capacity, we construct **LogicTale**, a benchmark comprising richly annotated stories, emphasizing causal reasoning, and visual logic interpretability. We establish comprehensive automatic and human evaluation protocols designed to measure both visual logic and perceptual

---

[*]Corresponding author.

quality. Experiments demonstrate that our approach significantly improves the narrative logic of generated visual stories. This work provides a foundational step towards modeling and enforcing visual logic in general image sequence and video generation tasks.

# 1 INTRODUCTION

Recent advances in generative models have enabled machines to produce high quality visual content from structured or unstructured inputs, such as text, sketches, or semantic layouts Zhou et al. (2025); Zhuang et al. (2024); Li et al. (2024). From image synthesis Liu et al. (2024a); Suo et al. (2025) to long-form video generation Dalal et al. (2025); Weijia Wu (2025); Guo et al. (2025), the ability to automatically construct visual narratives has opened exciting opportunities in creative domains ranging from illustration and education to filmmaking and simulation. Despite advances in high-fidelity content generation, ensuring that visual sequences evolve in a logically coherent manner that aligns with human expectations continues to be a key challenge.

Story visualization is a challenge task in visual sequence generation, where the goal is to generate a sequence of images that together depict a given narrative. Compared to single-image generation, this task requires not only high-quality visual outputs but also coherent storytelling across multiple images. Prior work has devoted substantial attention to improving visual quality and entity consistency Maharana et al. (2022); Pan et al. (2022); Rahman et al. (2022); Zhou et al. (2024); Tewel et al. (2024); Singh et al. (2025); Liu et al. (2025), which neglects narrative interpretability, resulting in sequences that resemble isolated image snapshots rather than causally connected visual narratives. Although significant progress has been made in textual logic modeling and planning Feng et al. (2023); Yao et al. (2019); Peng et al. (2022), there remains a substantial gap between textual and visual expression.

To address these challenges, we formally introduce the concept of **visual logic**. Visual logic refers to whether the progression of visual content across time or spatial layout forms an interpretable, causally sound, and semantically plausible experience for the viewer. As shown in Figure 1, failures related to visual logic in story visualization can manifest in many forms: abrupt object state changes without explanation, inconsistencies, emotionless or contradictory character behaviors. Building on this, we propose a structured generation framework **LogiStory**, which explicitly models visual logic through two components. First, we introduce a **Logic-Aware Multi-agent System** that transforms the input narrative into structured visual representations, including character definitions, object attributes, scene layouts, and panel-wise events. Second, we design a **Visual Logic Enhancement Module** that reinforces consistency through a Global Causal Verifier (which builds action-state graphs over the full story) and a Local Causal Monitor (which simulates human step-by-step comprehension during generation).

To support the development and evaluation of visual logic in generative models, we construct a new benchmark, **LogicTale**, featuring causal annotations, action-state flows, and panel-level story breakdowns. We further design visual logic-aware evaluation metrics that assess narrative coherence and interpretability beyond surface-level visual quality. In summary, the contribution of this paper includes:

(1) We define the concept of **visual logic** as a critical yet under-addressed dimension of generative modeling, and instantiate it through the task of multi-image story visualization.

(2) We propose **LogiStory**, a novel logic-aware framework for multi-image story visualization. LogiStory reframes storytelling as a logic-aware reasoning problem rather than isolated image synthesis, bridging structured narrative understanding with visual generation and explicitly strengthening story-level logic.

(3) We introduce **LogicTale**, a new benchmark dataset with annotated causal chains and structured visual representations, along with novel evaluation metrics that target visual logic consistency and narrative interpretability in visual sequences generation tasks.

# 2 RELATED WORK

## 2.1 STORY VISUALIZATION

Story visualization aims to generate a sequence of images that visually depict the progression of a narrative. Early work in this domain relied on GAN-based models Li et al. (2019); Feng et al. (2023), which generated one image per sentence with limited consideration for global coherence. The introduction of diffusion models significantly improved visual fidelity and control, enabling methods Zhou et al. (2024); Tewel et al. (2024); Shen & Elhoseiny (2025); Wang et al. (2024) to maintain character consistency across frames via identity conditioning. More recently, large language models have driven the development of fully automated pipelines. Agent-based systems Hu et al. (2024); Xu et al. (2025); Yang et al. (2024) perform explicit story planning to guide image generation, while multimodal large language models (MLLMs) such as GPT-4o OpenAI (2024) and Gemini 2.0 Flash DeepMind (2024) show strong performance in short-form comic generation. Despite these advances, existing methods often lack mechanisms for modeling and enforcing narrative logic across sequences. Our work addresses this gap by introducing visual logic as a core objective, and proposing the LogiStory framework to enhance the interpretability of story visualization.

## 2.2 CAUSAL REASONING IN TEXT AND VISUAL GENERATION

Maintaining causal coherence is critical for generating content that is logically understandable and narratively complete. In text-based story generation, early models often relied on superficial event orderings and struggled to capture causal dependencies between events Yamin et al. (2024). To address this, recent approaches Huot et al. (2024); Xi et al. (2025) incorporate structured planning, common sense inference, or intermediate causal representations such as event graphs, enabling models to better model cause-effect relations and produce more coherent narratives. In the visual domain, causal reasoning poses greater challenges due to the need for temporal consistency and interpretable transitions across images or video frames. Early story visualization methods Li et al. (2019); Liu et al. (2024a) lacked explicit mechanisms for cross-frame logic, while more recent approaches leverage character tracking Wu et al. (2024), structured representations Singh et al. (2025), or diffusion-based conditioning Zhou et al. (2024); Tewel et al. (2024) to maintain visual continuity. However, most of these methods focus on appearance or motion consistency rather than story readability and interpretability. Our work addresses this limitation by explicitly modeling visual logic, defined as the perceptual and causal coherence across a visual narrative, and introducing a generation framework that integrates structured planning with causal reasoning to ensure the intended story is both interpretable and logically sound.

## 2.3 EVALUATION OF VISUAL SEQUENCES

Evaluation of Visual Sequences remains a largely open problem, especially with regard to assessing narrative logic. Existing evaluations for visual storytelling often rely on standard image-level metrics such as FID for realism Heusel et al. (2018), CLIPScore for semantic alignment Radford et al. (2021), or DINO-based embedding distances for visual consistency Oquab et al. (2024). Alternatively, some works adopt generic benchmarks designed for image generation Huang et al. (2023); Ku et al. (2024); Wiles et al. (2024); Cho et al. (2024). While effective for assessing individual image quality, these approaches largely overlook inter-frame relationships, failing to evaluate whether the image sequence conveys a coherent narrative or maintains causal and temporal dependencies. In video generation, several benchmark suites have been proposed Liu et al. (2024b); Huang et al. (2024a;b), introducing a set of standardized metrics that assess frame quality, temporal alignment, and object continuity. However, even such benchmarks lack explicit evaluation of narrative-level logic or causal flow across scenes. To our knowledge, while VinaBench Gao et al. (2025) incorporates commonsense links and visual consistency, we are not aware of an existing benchmark that explicitly evaluates narrative-level logical coherence in visual story sequences, which motivates our emphasis on visual logic and causal reasoning.

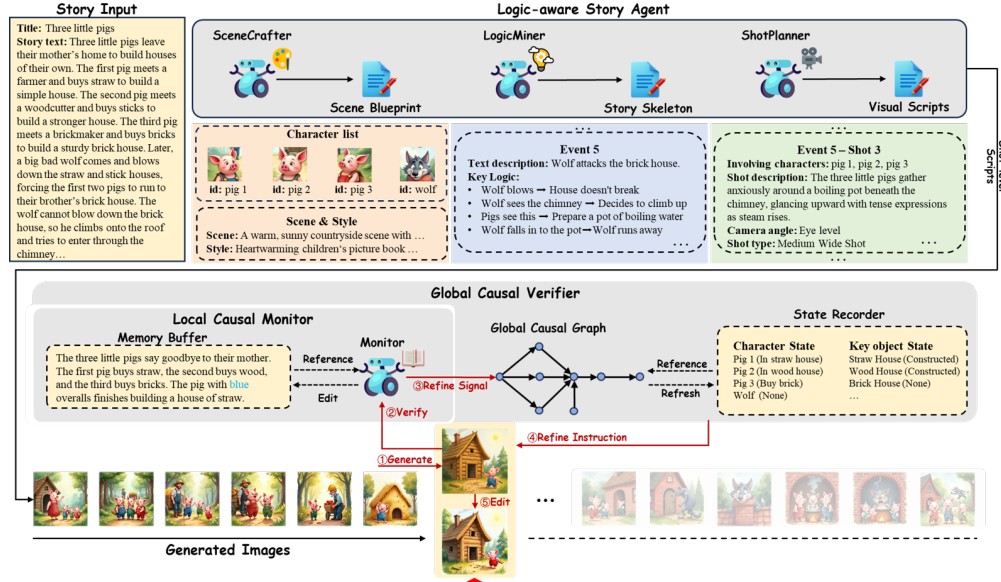

Figure 2: **Overview of LogiStory framework**. Given an input story, our system first applies a multi-agent story planner to decompose the story into structured panels with detailed scripts. In the generation process, the Local Causal Monitor simulates a reader's linear understanding by evaluating each frame for inconsistencies and generating refinement signals. Then, the Global Causal Verifier applies the causal graph to produce concrete refinement instructions to correct errors and maintain narrative flow.

## 3 METHOD

### 3.1 TASK FORMULATION

Given a text-form story $\mathcal{S}$, the goal is to generate a sequence of $T$ images $\mathcal{I} = \{I_1, I_2, \ldots, I_T\}$ such that the sequence visually depicts the storyline in a coherent, interpretable and logically consistent manner. Unlike traditional text-to-image generation, this task requires maintaining inter-frame dependencies across multiple dimensions, including: **(1) Instance consistency**: maintaining identity, appearance, and position of recurring instances. **(2) Narrative causality**: ensuring that actions and consequences follow a coherent and plausible causal chain. **(3) Story readability**: clearly conveying the intended narrative via the generated image sequence.

### 3.2 LOGISTORY FRAMEWORK

We propose **LogiStory** shown in Figure 2, including two components: **(1) Logic-Aware Multi-agent System** constructs structured intermediate representations from the input story. **(2) Visual Logic Enhancement Module** guides the synthesis and refinement of each image by incorporating both the story structure and multi-level visual logic verification. The detailed implementation(e.g., prompt templates) is provided in the Appendix D.

#### 3.2.1 LOGIC-AWARE MULTI-AGENT SYSTEM

The **Logic-Aware Multi-agent** transforms an input story into an interpretable intermediate representation. This system consists of three collaborative agents that work in stages: **(1) SceneCrafter**, an entity and context definition agent, **(2) LogicMiner**, a key event extraction agent, and **(3) Shot-Planner**, a shot-level planning agent. Together, they decompose the input narrative into structured components that guide the visual generation pipeline.

**SceneCrafter.** Given a story text $S$, the agent extracts and defines a set of semantic entities for consistent visual generation:

$$\mathcal{E} = \mathcal{F}_{\text{craft}}(S),$$

where $\mathcal{F}_{\text{craft}}(\cdot)$ defines attributes of entities in the story. The entity set $\mathcal{E}$ comprises characters $\mathcal{C} = \{c_1, \ldots, c_K\}$, objects $\mathcal{O} = \{o_1, \ldots, o_M\}$, and scenes $\mathcal{S} = \{s_1, \ldots, s_N\}$. These structured definitions form a visual grounding vocabulary, reused across all panels to ensure stable and consistent representation throughout the story, ensuring stable representation of characters and objects across different scenes and supports downstream modules in maintaining entity consistency and contextual coherence.

**LogicMiner.** To guide visual composition, we extract key narrative events defined as tuples $k_j = (\text{actor}, \text{action}, \text{target}, \text{result})$, capturing causally relevant or state-changing moments in the story. Given the input text $S$, we apply a large language model to directly extract these structured events. The function can be formalized as:

$$\mathcal{K} = \{k_1, \ldots, k_J\} = \mathcal{F}_{\text{mine}}(S, \mathcal{E}),$$

where $\mathcal{F}_{\text{mine}}(\cdot)$ identifies both explicitly stated and implicitly implied state transitions critical for visual grounding. This dual capability ensures that both direct narrative cues and essential inferred changes are captured for visual grounding. For example, from "the crow put pebbles into the cup," LogicMiner infers the rising water level. These events form the causal backbone, guiding subsequent shot planning and scene arrangement.

**ShotPlanner.** Given key events $\mathcal{K}$ and entities $\mathcal{E}$, the shot planning agent organizes them into a sequence of panel specifications. The function can be formalized as:

$$\mathcal{P} = \{p_1, \ldots, p_T\} = \mathcal{F}_{\text{shot}}(\mathcal{K}, \mathcal{E}),$$

where $\mathcal{F}_{\text{shot}}(\cdot)$ designs detailed shot specifications, including characters, actions, objects, spatial relations, scenes, and camera parameters. ShotPlanner incorporates visual storytelling conventions (e.g., pacing, framing, perspective) to control narrative rhythm, emphasize key events, and ensure visually coherent, engaging sequences, ensuring the resulting image sequence not only captures the intended events but also presents them in a visually engaging and narratively coherent manner.

### 3.2.2 Visual Logic Enhancement Module

Although structured planning provides high-level narrative scaffolding, ensuring that the generated image sequence aligns with human-interpretable logic requires deeper modeling of visual logic. We propose a **Visual Logic Enhancement Module** to bridge the gap between semantic planning and visual realization. This module comprises two complementary components as follows.

**Local Causal Monitor.** To model the linear comprehension process of human readers, we introduce a Local Causal Monitor that evaluates each panel generation step by step. We maintain a text-form causal memory buffer $\mathcal{M}_{t-1}$ consisting of state snapshots and actions from $\{p_1, \ldots, p_{t-1}\}$. Given the current panel image $I_t$ and its local context $\mathcal{M}_{t-1}$, we simulate a human reading path by checking whether $p_t$ remains narratively plausible and coherent with respect to $\mathcal{M}_{t-1}$, rather than enforcing deterministic prediction. Using MLLMs to simulate the reader's reasoning process, we define a causal plausibility score $\psi_t$ as:

---

**Algorithm 1** Visual Enhancement Module

**Input:** Panel list $\mathcal{P} = \{p_1, \ldots, p_T\}$, story text $S$, key logic $\mathcal{K} = \{k_1, \ldots, k_J\}$
**Output:** Image sequence $\{I_1, \ldots, I_T\}$
Construct causal graph $\mathcal{G}_{\text{causal}}$ from $S$ and $\mathcal{K}$  Initialize memory buffer $M_0$
**for** $t \leftarrow 1$ **to** $T$ **do**
    Generate draft image $I_t$ for panel $p_t$  Compute causal score $\psi_t = C_p(I_t \mid M_{t-1})$
    **if** $\psi_t < \tau_1$ **then**
        Regenerate $I_t$
    **else**
        **if** $\psi_t \in [\tau_1, \tau_2)$ **then**
            Invoke Global Causal Verifier  Edit $I_t$ using visual tools
        **else**
            Accept $I_t$
    Update $M_t \leftarrow M_{t-1} \cup \{Caption(I_t)\}$
**return** $\{I_1, \ldots, I_T\}$

---

$$\psi_t = C_p\left(I_t \mid \mathcal{M}_{t-1}\right),$$

where $C_p$ evaluates the degree to which the depicted state transitions are consistent with or reasonably extend the accumulated context. This allows the monitor to accommodate both expected progressions and surprising developments, as long as they preserve overall narrative logic and do not contradict prior states.

**Global Causal Verifier.** Based on the extracted key events $\mathcal{K}$ and story text $S$, we construct a directed causal graph $\mathcal{G}_{\text{causal}}$ representing the narrative's logical backbone, where nodes represent key

states (e.g., character status, object conditions), and edges represent causal or temporal dependencies inferred from the event semantics. A state recorder tracks the current states of all instances, updating dynamically as the story progresses to provide a reference for verification. For each generated image $p_t$, the verifier checks whether the visualized state transitions match the expected causal links in the graph. Specifically, for an action $a_t$, we define its pre-condition state $S_t^{\text{pre}}$ and post-condition state $S_t^{\text{post}}$:

$$S_t^{\text{pre}} \xrightarrow{a_t} S_t^{\text{post}}.$$

These transitions are validated against the causal graph to ensure consistency with the established narrative chains:

$$\forall t, \quad (\mathcal{S}_t^{\text{pre}}, \mathcal{S}_t^{\text{post}}) \in \text{Paths}(\mathcal{G}_{\text{causal}}).$$

Any inconsistency is flagged and sent to image editing tools for correction.

**Image Refinement.** To ensure logical coherence during generation, each panel $p_t$ is immediately evaluated upon creation. Given the logic confidence score $\psi_t \in [0, 1]$ using the Local Causal Monitor. Refinement decisions are made according to Algorithm 1. The thresholds $\tau_1 = 0.4$ and $\tau_2 = 0.7$ are empirically set based on preliminary experiments. $\tau_1$ marks the boundary below which the generated image is considered unrelated to the narrative, while $\tau_2$ indicates sufficient alignment with the intended meaning. If refinement is needed ($\tau_1 \leq \psi < \tau_2$), we invoke the Global Causal Verifier, which maintains a structured causal graph of actions and states. It provides explicit revision instructions, guiding targeted editing via image editing tools.

## 4 LOGICTALE BENCHMARK

Since existing story visualization benchmarks lack direct evaluation of **story-level logic**, we design a dataset and evaluation suite, **LogicTale**, to explicitly assess visual logic in multi-image narratives and to facilitate the development of logic-aware visual content generation. The detailed dataset composition and evaluation protocol definition are provided in the Appendix C.

### 4.1 DATASET CONSTRUCTION

**Dataset Composition.** The dataset contains 60 meticulously annotated stories. To ensure both diversity and generalization, we include a balanced mix of well-known classic stories and human-authored original stories, in a 3:2 ratio. The inclusion of well-known classic stories ensures the fundamental quality and reliability of the dataset, while original stories introduce unseen structures that posing greater challenges for the task. Stories are labeled as *easy*, *medium*, or *hard* based on visual reasoning difficulty.

**Data Annotation.** Each story contains the following components: (1) A story title and its source. (2) The full narrative story text. (3) A character list specifying key and supporting entities. (4) A set of visual logic chains annotated as tuples (action, result, weight), where each tuple represents a causally important event, and $\sum_i \text{weight}_i = 1$ to normalize importance. (5) A difficulty label reflecting the expected complexity in visual story modeling.

**Dataset Scale.** Regarding **evaluation scale**, our dataset is **comparable or larger** than prior works: *StoryDiffusion* (**20** short single-character stories) Zhou et al. (2024), *ConsiStory* (**20** hand-crafted scenes) Tewel et al. (2024), *MM-StoryAgent* (**100** LLM-generated stories) Xu et al. (2025), *MovieAgent* (**12** authored examples) Weijia Wu (2025), and *ViStoryBench* (**80** test cases) Zhuang et al. (2025). Thus, LogicTale's 60 richly annotated, multi-character stories with varied difficulty provide a **solid, representative benchmark**.

### 4.2 EVALUATION PROTOCOL

To evaluate the performance, we consider two complementary dimensions: **visual logic** and **perceptual quality**. Together, these evaluation components form a comprehensive benchmark that captures both the logical interpretability and visual quality of generative systems in visual storytelling.

**Visual Logic Evaluation.** We evaluate visual logic from three aspects. **(1) Instance Consistency** metrics evaluate whether key elements are preserved across frames via MLLMs. Character consistency examines whether character appearance, clothing, and identity remain visually stable. Object

consistency checks whether the appearance and transformation of key objects follow a plausible state evolution. Scene consistency ensures that background environments remain locally coherent unless disrupted by explicit narrative cues. **(2) Narrative Causality** depends on whether the generated images accurately express the annotated key events. Each event $e_i = (\text{action}, \text{result}, \text{weight})$ is assigned a quality score based on its clarity, coherence, and plausibility. The overall event-based score is computed as:

$$\text{CausalScore} = \sum_i \text{EventScore}(e_i) \cdot \text{weight}_i.$$

**(3) Story Readability** represents how well the generated image sequence conveys the intended narrative. To do this, we first apply a captioning model (e.g., BLIP-2) to produce a textual description for the entire image sequence. We then provide an LLM with two inputs: the story's character list and the generated caption. The model is prompted to infer the underlying story based on this information. Finally, we compute semantic similarity between the inferred story and the original ground-truth narrative to assess global story alignment.

**Perceptual Quality Evaluation.** Perceptual quality focus on the qualities of image presentation. **(1) Aesthetic Quality** is assessed using HPSv2 to evaluate the visual appeal of the image sequence. **(2) Style Consistency** examines whether the entire sequence maintains a coherent visual rendering style, estimated via DINOv2 embedding distance. **(3) Character Expressiveness** evaluates how well the emotions, poses, and gestures of characters align with the narrative events and is rated by MLLMs.

## 5 EXPERIMENTS

### 5.1 EXPERIMENT SETTING

**Dataset.** We conduct experiments on the proposed LogicTale dataset, providing a diverse testbed for evaluating visual narrative generation. For fairness and generality, we also compare against datasets used in prior works such as *ViStoryBench* and *StoryDiffusion*, and report corresponding evaluations in the Appendix B.

**Metrics**. We evaluate performance using both automatic and human assessments. **Automatic evaluation** follows LogicTale evaluation protocol. **Human evaluation** is designed to closely align with our automatic evaluation protocol. To rigorously assess story-level logical coherence, we include three dimensions: *Instance Consistency*, *Narrative Causality*, and *Story Readability*. This design enables human judgment to provide a fine-grained evaluation of the core narrative logic aspects targeted in our framework. In contrast, perceptual quality is evaluated using a single dimension, *Aesthetic Appeal*, which captures overall visual attractiveness and stylistic coherence. We adopt this simplified form because perceptual quality is not the central focus of this work and to reduce annotation cost. For each test case, annotators are presented with the full story text together with the generated image sequence. The detailed implementations are provided in the Appendix C.

**LogiStory Settings.** The agent module leverages DeepSeek-V3 as the base LLM. The image generation module utilizes Flux for initial panel synthesis. For image refinement and editing, we integrate inpainting models and MLLMs, including GPT-image-1 and Gemini 2.0 Flash, to support targeted edits guided by the verifier's feedback.

**Baselines.** We compare our method with both closed-source and open-source baselines. For closed models, we evaluate the most capable publicly accessible systems to date: **Nano Banana** Google DeepMind (2025), **Gemini 2.0 Flash** DeepMind (2024) and **GPT-4o+GPT-image-1** OpenAI (2024), both of which support end-to-end generation from story text to panel planning and visual sequence rendering. For open-source baselines, we select models capable of open-ended and general-purpose story visualization, as opposed to methods fine-tuned on specific datasets (e.g., StoryGPT-V Shen & Elhoseiny (2025)). We employ the end-to-end agent-based framework **MM-StoryAgent** Xu et al. (2025). While existing works such as **StoryDiffusion** Zhou et al. (2024) and **ConsiStory** Tewel et al. (2024) lack built-in story planning modules. To ensure fair comparison, we pair these methods with **DeepSeek-R1** DeepSeek-AI (2025) as a planning backbone to generate panel-level scene descriptions. We also evaluate the performance of standard generative models including **SDXL** Podell et al. (2023) and **Flux** Labs (2024) under the same setup.

Table 1: Performance of Automatic metric on story visualization methods. "ICons.", "NCausal.", "SRead.", "AesthQ.", "SCons.", "CExpr.", "App." refer to "Instance Consistency", "Narrative Causality", "Story Readability", "Aesthetic Quality", "Style Consistency", "Character Expressiveness", "Aesthetic Appeal". The first rank is highlighted in **bold**, while the second rank is underlined.

| Method | Visual Logic | | | Perceptual Quality | | | User study | | | |
|---|---|---|---|---|---|---|---|---|---|---|
| | ICons. | NCausal. | SRead. | AesthQ. | SCons. | CExpr. | ICons. | NCausal. | SRead. | App. |
| DeepSeek-R1 + SDXL | 2.17 | 2.02 | 0.5675 | 0.2899 | 0.7747 | 2.13 | 2.33 | 1.86 | 2.13 | 3.0 |
| DeepSeek-R1 + Flux | 3.87 | 3.43 | 0.6872 | 0.3047 | 0.8329 | 4.20 | 3.53 | 3.25 | 3.20 | 3.8 |
| DeepSeek-R1 + StoryDiff | 3.07 | 2.06 | 0.5409 | 0.2909 | 0.8276 | 2.07 | 2.70 | 2.12 | 2.35 | 2.6 |
| DeepSeek-R1 + ConSistory | 3.23 | 1.98 | 0.5623 | 0.2939 | 0.8129 | 2.26 | 2.87 | 1.73 | 1.87 | 2.4 |
| MM-StoryAgent | 3.03 | 2.63 | 0.5787 | 0.2942 | 0.8023 | 2.54 | 2.67 | 3.04 | 2.43 | 2.6 |
| Gemini 2.0 flash | 3.62 | 3.88 | 0.6510 | 0.2547 | 0.8197 | 3.48 | 3.7 | 3.65 | 3.58 | 2.8 |
| Nano Banana | 4.20 | 4.08 | 0.7440 | **0.3102** | **0.9034** | 4.26 | 4.20 | 3.96 | 3.75 | 4.4 |
| GPT-4o + GPT-image-1 | **4.67** | 3.96 | 0.7622 | 0.2829 | 0.8984 | 4.15 | **4.38** | 3.78 | 3.65 | **4.6** |
| Ours | 4.23 | **4.45** | **0.8267** | 0.3088 | 0.8572 | **4.32** | 4.25 | **4.26** | **3.83** | 4.2 |

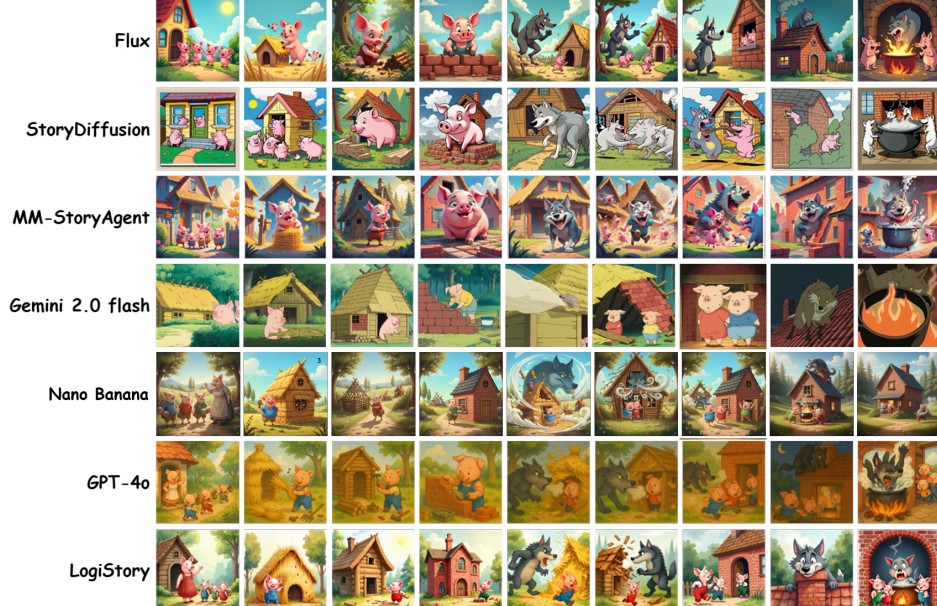

Figure 3: **Quality analysis of generated stories across different methods.** Representative key scenes are shown here. Complete sequences can be found in the Appendix A.

## 5.2 PERFORMANCE COMPARISONS AND ANALYSIS

### 5.2.1 AUTOMATIC EVALUATION

The results are shown in Table 1. Our approach achieves the highest score in Narrative Causality, Story Readability, Aesthetic Quality and Character Expressiveness. In terms of Instance Consistency and Style Consistency, our method slightly trails behind GPT-Image-1, which benefits from its strong ability to edit reference images. Nevertheless, our method shows competitive performance in this regard. Figure 3 presents a qualitative comparison of generated results on the story *Three Little Pigs*. Open-source methods, including **Flux** (basic diffusion model), **StoryDiffusion** (character-consistent generation model), and **MM-StoryAgent** (agent-based system), all exhibit common errors such as attribute mismatches and inconsistent visual elements, resulting in low narrative clarity and weak interpretability. **Nano Banana** and **Gemini 2.0 Flash** demonstrates a relatively clear planning of story rhythm. However, it still suffers from instance omissions and attribute errors. Similarly, **GPT-4o+GPT-image-1** shows attribute confusion issues, such as mixing the clothing of the three pigs. In contrast, **LogiStory** generates story sequences with clearer logic, stronger narrative readability, and higher consistency between characters, actions, and scenes, effectively addressing the challenges of visual logic in multi-image story visualization.

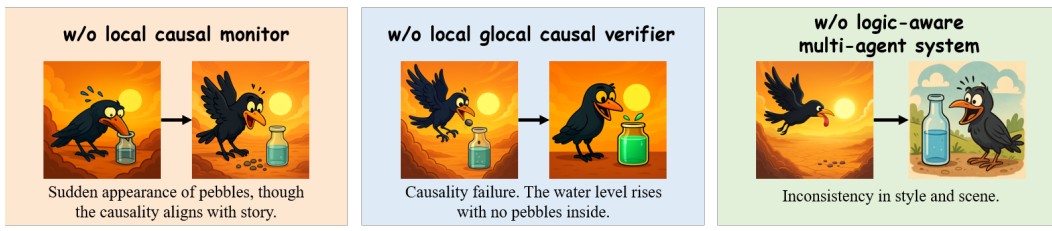

Figure 4: **Ablation study on key components of LogiStory.** Qualitative comparisons on representative examples demonstrate the impact of each module.

### 5.2.2 HUMAN RATING

To comprehensively assess subjective quality, we conducted a user study with 30 participants. Each was shown randomly shuffled outputs from all methods for the same story, without knowing their identities to avoid bias. The results in Table 1 show that our method outperforms others on the three logic-focused dimensions: **Instance Consistency**, **Narrative Causality**, and **Story Readability**. It also achieves high scores in **Aesthetic Appeal**, indicating both logical clarity and visual attractiveness. We further measured alignment between user study and automatic evaluation using **Pearson correlation**, yielding strong results (Instance Consistency: **0.959**, Narrative Causality: **0.978**, Story Readability: **0.909**, Perceptual Quality: 0.695). The lower correlation for perceptual quality is mainly due to stylistic homogeneity among SDXL-based methods such as StoryDiffusion and ConsisStory, which reduces discriminative power.

### 5.3 ABLATION STUDY

To better understand the contribution of each module in our framework, we conduct ablation experiments on **logic-aware multi-agent system** and **visual logic enhancement module**. Since our work primarily focuses on enhancing visual logic, we restrict the evaluation to the visual logic metrics in our benchmark. Qualitative results are shown in Figure 4

**Effect of Planning Methods:** To evaluate the effectiveness of our multi-agent planning module, we conduct an ablation study by replacing it with an LLM-based planner, where the LLM directly outputs a sequence of actions based on the story description. All other components of the **LogiStory** framework remain unchanged. As shown in Table 2, compared to directly prompting large models such as DeepSeek-V3 DeepSeek-AI (2024), DeepSeek-R1 DeepSeek-AI (2025), or Qwen2.5-72B Team (2024), the inclusion of our multi-agent story planning system significantly improves performance across all visual logic metrics. This demonstrates the effectiveness of decomposing the narrative into structured roles and key events, which provides strong priors for coherent image sequence generation.

**Effect of Global Causal Verifier:** To evaluate the role of the Global Causal Verifier in ensuring story-level causal coherence, we conduct an ablation study where the original verifier is replaced by an LLM-based baseline. Specifically, we remove the Global Causal Verifier module and instead use an LLM to directly generate refinement instructions based solely on the story text without structured causal reasoning. As shown in Table 3, the integration of the Global Causal Verifier leads to notable gains in **Narrative Causality** scores. We attribute this to its ability to explicitly monitor and enforce state-action-result chains throughout the story, ensuring that critical logical transitions are preserved in the visual narrative.

**Effect of Local Causal Monitor:** We further assess the contribution of the Local Causal Monitor by removing this module from the pipeline. In this ablation setting, the system bypasses the local monitor and instead directly evaluates the alignment between each generated image and its corresponding shot-level script using a simple matching score. As shown in Table 3, adding the Local Causal Monitor brings consistent improvements in **Story Readability**. By simulating the human reading process and performing incremental causal validation during generation, this module enhances the overall clarity and interpretability of the story.

Table 2: Ablation study on story scripts planning methods.

| Planning Method | ICons. | NCausal. | SRead. |
|---|---|---|---|
| DeepSeek-V3 | 3.92 | 3.64 | 0.7233 |
| DeepSeek-R1 | 3.87 | 3.86 | 0.7159 |
| Qwen2.5-72B | 3.66 | 3.52 | 0.6614 |
| Ours | **4.23** | **4.45** | **0.8267** |

Table 3: Ablation study on visual logic enhancement module.

| Method | ICons. | NCausal. | SRead. |
|---|---|---|---|
| Ours (w/ none) | 4.16 | 3.93 | 0.6527 |
| Ours (w/ global) | 4.12 | 4.27 | 0.7283 |
| Ours (w/ local) | 4.24 | 4.10 | 0.7732 |
| Ours (w/ both) | **4.23** | **4.45** | **0.8267** |

## 5.4 COMPLEX CASE AND FAILURE ANALYSIS

In this subsection, we present a representative *hard-level* case featuring **multi-task narrative structure**, flashback usage, and **subtle emotional implications**. We provide the fully rendered **causal graph** alongside **key panel comparisons** for illustration. As shown in Figure 5, the framework constructed **causal graph (b)** effectively decomposes the core logical dependencies and developmental relationships within the story.

For such complex scenarios, we observe that the primary bottleneck lies **on the generation side rather than the planning stage**. Our framework demonstrates **higher character consistency** and **better narrative alignment** compared with representative baselines such as *StoryDiffusion* and *GPT-4o + GPT-image-1*. However, certain challenges remain: for instance, **visual emotional expression may be insufficiently delicate**, and **event-level coherence can occasionally degrade**. This is partly due to the inherent difficulty of visually presenting fine-grained emotional trajectories, especially when narrative intent relies strongly on subtle affective cues.

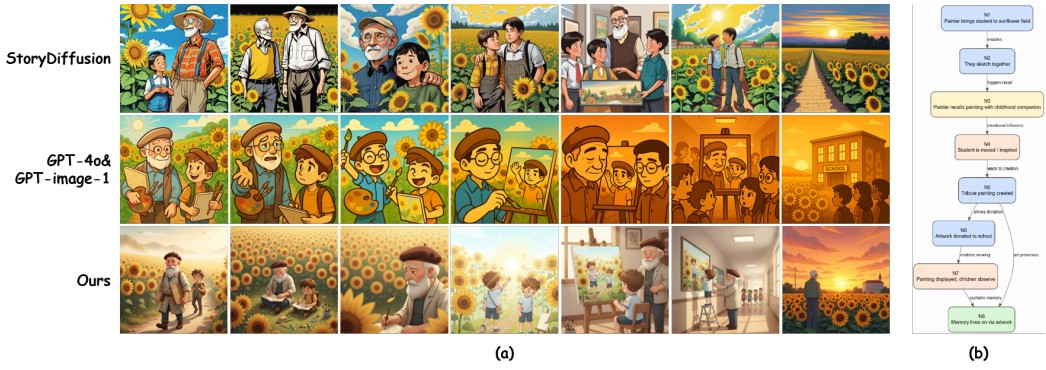

Figure 5: **Quality analysis on Complex Case.** *Memory Lasts: An old painter takes his student to a distant sunflower field, once shared with his younger brother before the war. As they paint, the narrative shifts briefly to a childhood flashback, two boys painting under the sunflower sky, one waving goodbye. Upon returning home, the student donates a commemorative painting to a local school, where it remains as the old painter's lasting memory.*

## 6 CONCLUSION

In this work, we introduce **visual logic** as a central objective in story visualization, tackling the under-explored challenge of ensuring causally coherent and semantically plausible storytelling across image sequences. We present **LogiStory**, a structured framework that combines multi-agent planning with causal reasoning, producing sequences that are both visually appealing and narratively coherent. To enable rigorous evaluation, we construct **LogicTale**, a benchmark with causal annotations, difficulty levels, and protocols for both automatic and human assessment. Experiments show that our approach surpasses existing baselines, especially in modeling complex visual causality. We believe this work lays a foundation for logic-aware generation and offers insights for advancing story visualization and video synthesis.

## 7 ETHICS STATEMENT

This work focuses on logic-aware story visualization framework LogiStory and the construction of the LogicTale dataset. The dataset is composed of publicly available narratives (e.g., classic literature, fables, and human-authored short stories) that do not contain private or sensitive information. All human annotations were collected from consenting participants, who were informed of the purpose of the study and compensated fairly. We ensured that annotators' personal data were not collected or stored, thus preserving privacy. The proposed framework, LogiStory, is a general-purpose research system and does not target harmful or sensitive applications. Nevertheless, as with other generative models, there is a risk of producing biased or culturally inappropriate content. To mitigate this, we include stories from diverse cultural backgrounds and explicitly annotate logical structures to encourage fairer and more interpretable evaluations. To support transparency and reproducibility, we plan to release both the dataset and implementation code in the near future, subject to proper documentation and licensing considerations.

## 8 REPRODUCIBILITY STATEMENT

We are committed to ensuring the reproducibility of our work. The implementation details of the **LogiStory** framework, including model configurations and pipeline design, are provided in Appendix D. The construction process of the **LogicTale** benchmark, along with annotation protocols and evaluation procedures, is described in Appendix C. We provide a subset of LogicTale in the supplementary material. These materials are intended to allow researchers to replicate both our framework and evaluation methodology.

## 9 ACKNOWLEDGMENTS

This work is supported by the National Natural Science Foundation of China (62436007).

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

## A    MORE QUALITATIVE RESULTS

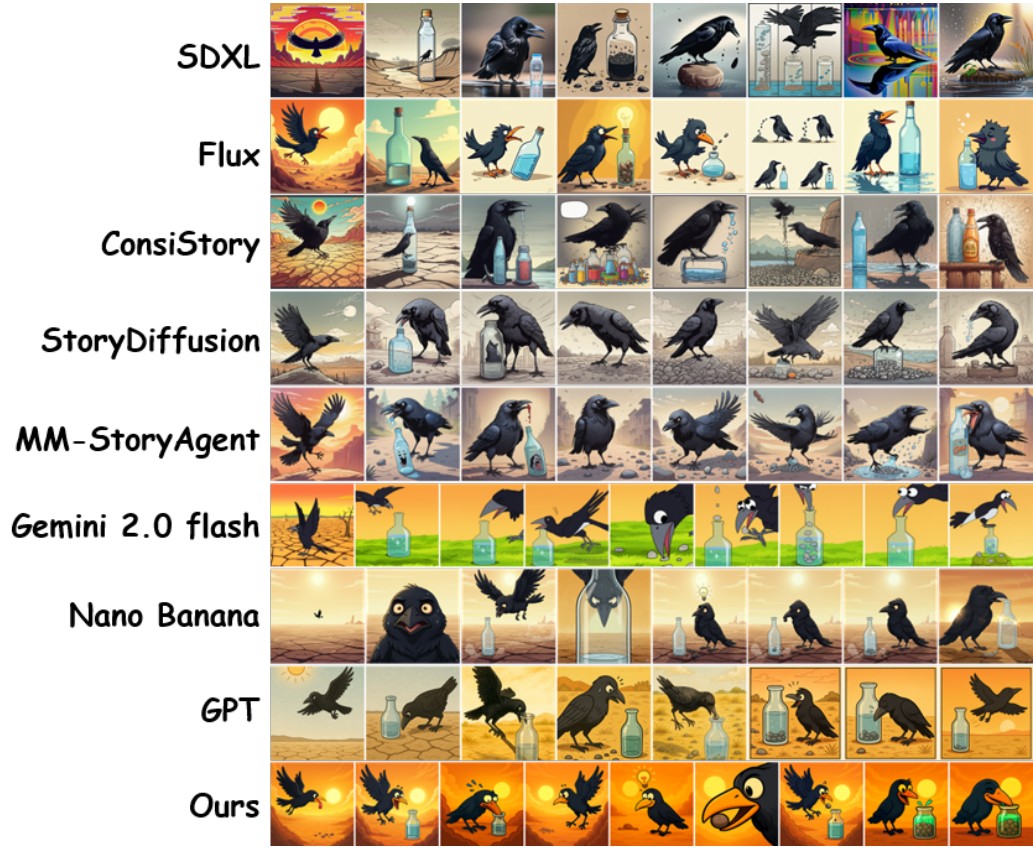

Figure 6: **Quality analysis of generated stories across different methods.** *The Crow and the Pitcher: Under the scorching sun, across a vast, parched land, a thirsty crow flies in search of water. It eventually spots a tall glass bottle with a small amount of water at the bottom. Unable to reach it with its beak, the crow notices small pebbles scattered on the ground. One by one, it picks up the pebbles and drops them into the bottle. As the stones pile up, the water level slowly rises. At last, the water reaches the top, and the crow happily quenches its thirst.*

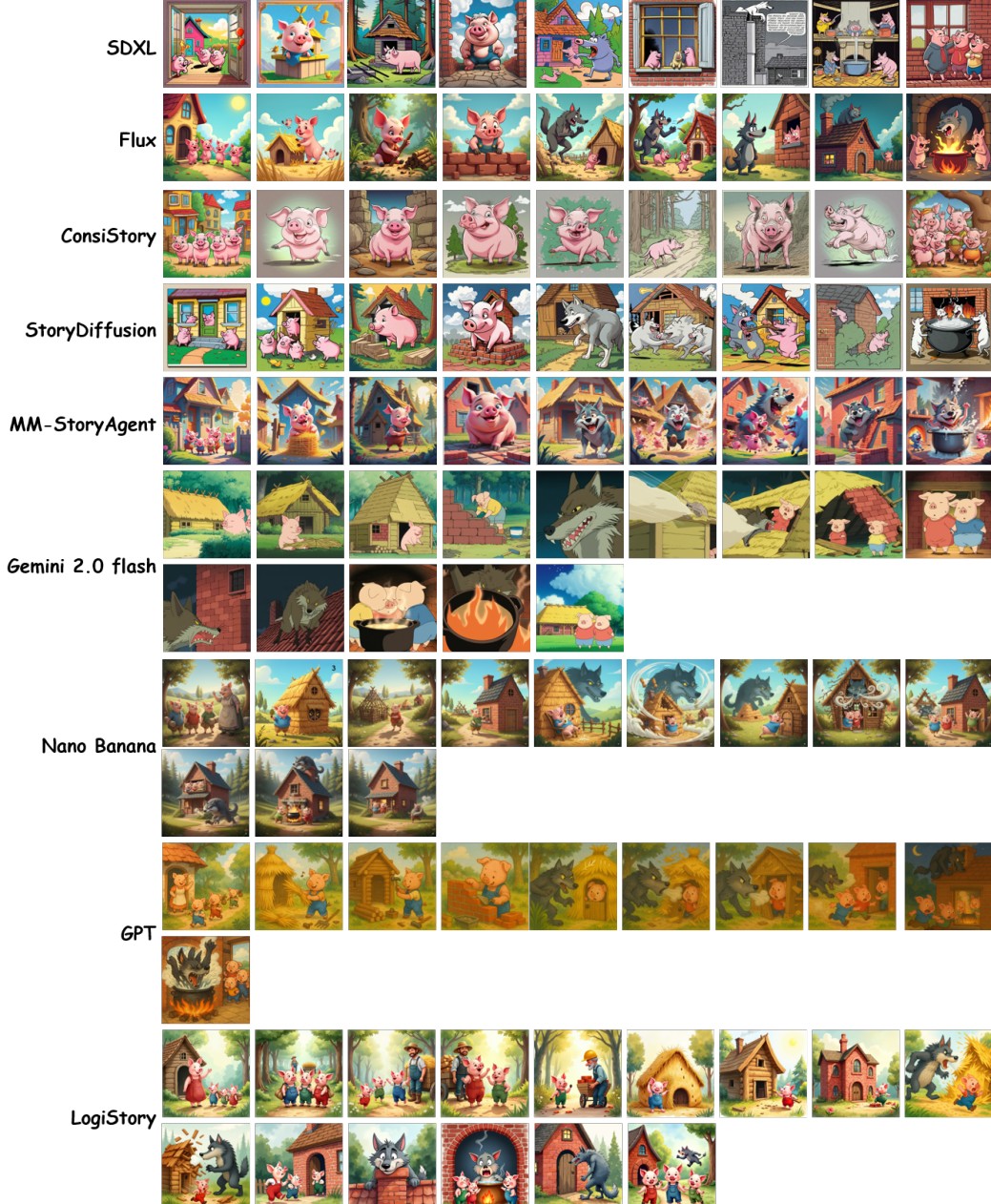

Figure 7: **Quality analysis of generated stories across different methods.** *Three Little Pigs: Three little pigs live their mother's house and each build a house: one of straw, one of sticks, and one of bricks. A hungry wolf comes and blows down the straw and stick houses, causing the first two pigs to flee to the brick house. The wolf tries to blow it down but fails. He then climbs the chimney, but the pigs boil a pot of water inside, and the wolf falls in and runs away burned. All three pigs live safely together in the brick house.*

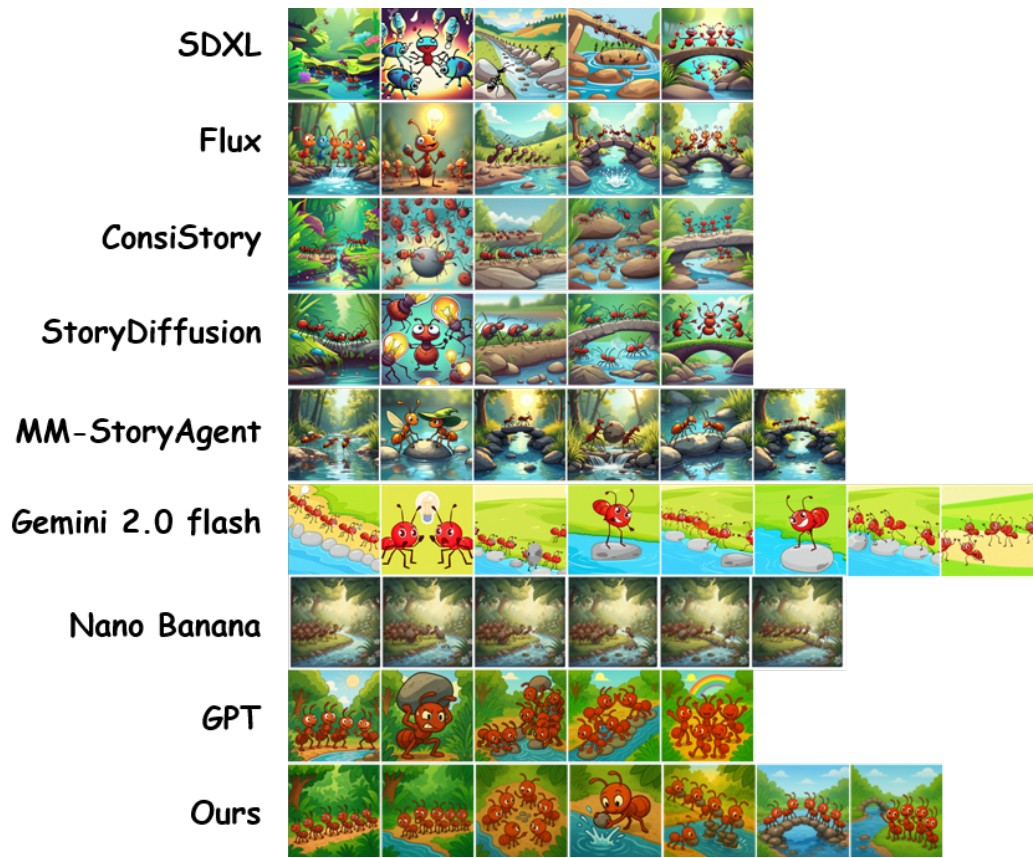

Figure 8: **Quality analysis of generated stories across different methods.** *The Pebble Bridge: A group of ants needs to cross a small stream. They drop pebbles into the water to form a bridge and successfully cross together.*

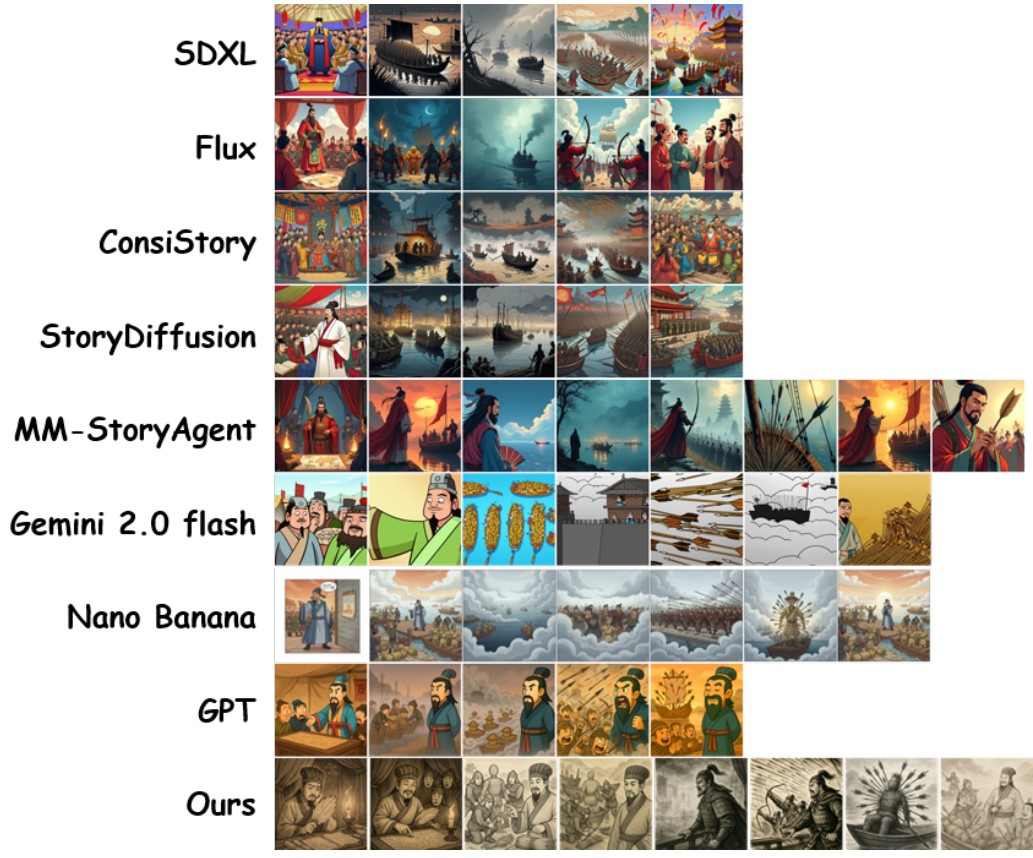

Figure 9: **Quality analysis of generated stories across different methods.** *Borrowing Arrows with Straw Boats: During a critical shortage of arrows, Zhuge Liang boldly promised to deliver 100,000 arrows within three days. To fulfill this promise, he ordered countless straw men to be placed on boats and, under the cover of thick fog, sailed toward Cao Cao's camp. Mistaking the figures for soldiers, the enemy forces unleashed volleys of arrows at the boats. When the boats returned, they were laden with arrows, allowing Zhuge Liang to fulfill his promise cleverly and without shedding a drop of blood.*

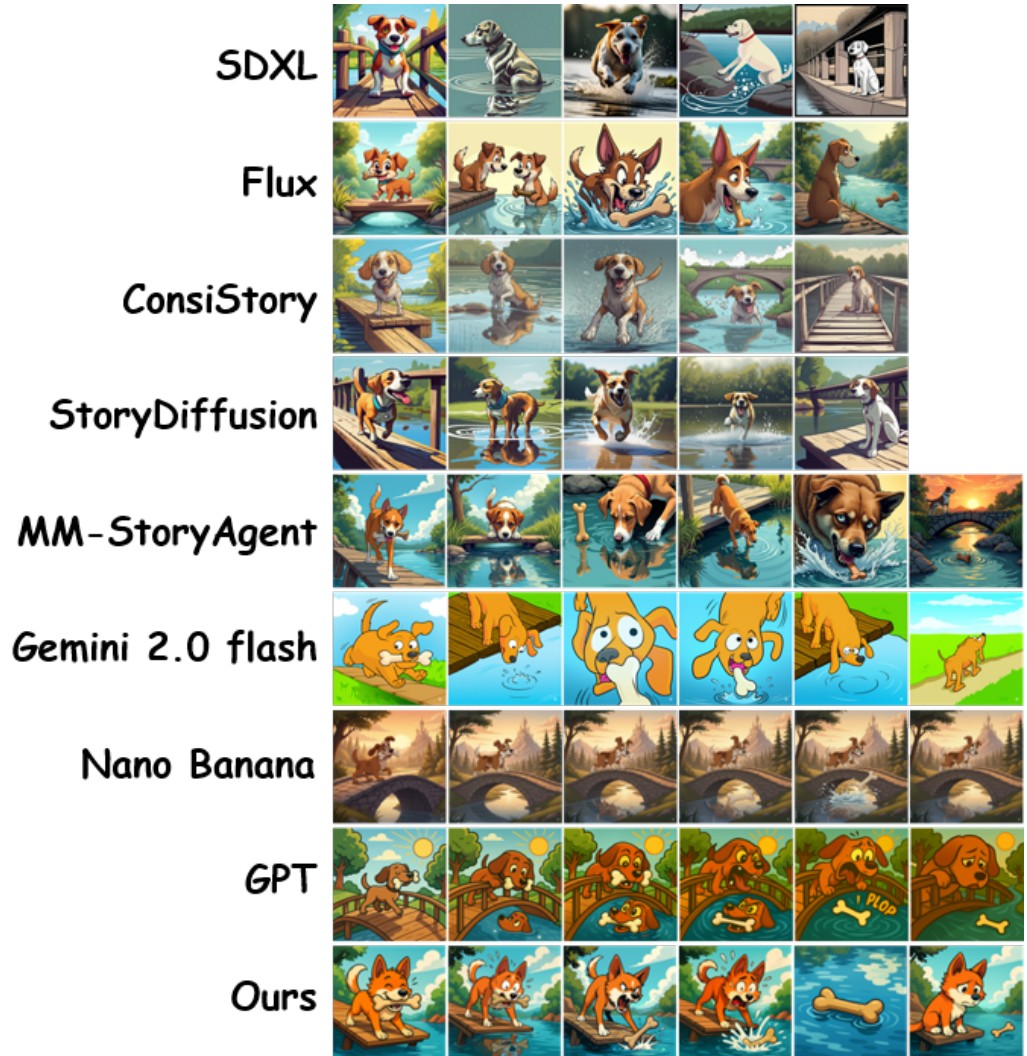

Figure 10: **Quality analysis of generated stories across different methods.** *The Dog and His Reflection: A dog crosses a bridge with a bone in its mouth. Looking into the water, he sees his own reflection and mistakes it for another dog with a bigger bone. He snaps at the reflection, dropping his bone into the river and losing it.*

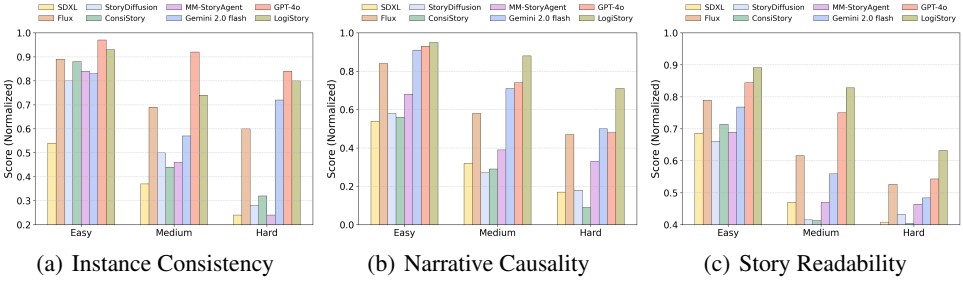

(a) Instance Consistency    (b) Narrative Causality    (c) Story Readability

Figure 11: Comparative evaluation of different methods across difficulty levels.

## B    MORE QUANTITATIVE RESULTS

### B.1    MORE METHODS PERFORMANCE ON LOGICTALE

To further demonstrate the robustness of our framework, we include results of additional baselines on the **LogicTale** benchmark. Table 4 reports the performance across six dimensions: *Instance Consistency (ICons.)*, *Narrative Causality (NCausal.)*, *Story Readability (SRead.)*, *Aesthetic Quality (AesthQ.)*, *Style Consistency (SCons.)*, and *Character Expressiveness (CExpr.)*.

Table 4: Performance comparison of additional baselines on the **LogicTale** benchmark.

| Method | ICons. ↑ | NCausal. ↑ | SRead. ↑ | AesthQ. ↑ | SCons. ↑ | CExpr. ↑ |
|---|---|---|---|---|---|---|
| StoryGen | 3.24 | 1.72 | 0.5366 | 0.2873 | 0.8162 | 1.92 |
| StoryDiffusion | 3.07 | 2.06 | 0.5409 | 0.2909 | 0.8276 | 2.07 |
| Story-Adapter | 3.65 | 2.23 | 0.5733 | 0.2952 | 0.8195 | 3.16 |
| **Ours** | **4.23** | **4.45** | **0.8267** | **0.3088** | **0.8572** | **4.32** |

As shown in Table 4, our method consistently outperforms all baselines across both **logical dimensions** (Instance Consistency, Narrative Causality, Story Readability) and **visual dimensions** (Aesthetic Quality, Style Consistency, Character Expressiveness). These results confirm that LogiStory achieves superior performance in balancing narrative logic with visual presentation, reinforcing its role as the first framework explicitly designed for logic-aware visual storytelling.

### B.2    ANALYSIS ON DIFFERENT DIFFICULTY LEVELS

A further analysis based on the difficulty levels shown in Figure 11 reveals that our method's advantage becomes more pronounced as the story complexity increases. This highlights the robustness of our approach in handling more intricate story structures.

### B.3    LOGISTORY WITH DIFFERENT BASE MODELS

To evaluate the adaptability of **LogiStory** to different backbone models, we tested the framework with smaller vision-language models: **Qwen2.5-VL-7B**, **InternVL 2.5-8B**, and **Qwen2.5-VL-32B**. For reference, we also include results with the larger **DeepSeek-V3** backbone. Results are summarized in Table 5.

Table 5: Performance of LogiStory with different backbone models.

| Backbone | ICons. ↑ | NCausal. ↑ | SRead. ↑ |
|---|---|---|---|
| Qwen2.5-VL-7B | 3.04 | 3.32 | 0.6132 |
| InternVL 2.5-8B | 3.21 | 3.24 | 0.6328 |
| Qwen2.5-VL-32B | 3.85 | 3.78 | 0.7244 |
| DeepSeek-V3 | **4.26** | **4.45** | **0.8267** |

As shown in Table 5, smaller backbones yield lower scores on logic-related metrics, but performance still surpasses baseline methods, confirming the framework's robustness.

A closer analysis identifies three factors behind the performance gap:

- **Shorter panel scripts**: Smaller models generate less detailed scene breakdowns, limiting narrative depth.
- **Weaker multi-agent communication**: Semantic information is more likely to be lost across stages, reducing coherence.
- **Homogeneous causal scores in Local Causal Monitor**: Smaller LLMs assign uniformly high scores, failing to trigger necessary refinements.

These findings highlight that while stronger backbones enhance story-level logic modeling, **LogiStory remains effective and competitive even with lightweight models**.

### B.4 EVALUATION ON OTHER DATASETS

We conducted additional evaluations on **ViStoryBench-Lite** to further assess the generalizability of our framework. Results are summarized in Table 6.

Table 6: Comparison on ViStoryBench-Lite.

| Method | CSD Self ↑ | CIDS Self ↑ | Alignment ↑ | OCCM ↑ | Inception ↑ | Aesthetics ↑ |
|---|---|---|---|---|---|---|
| GPT-4o | 68.5 | **73.1** | 89.3 | 93.4 | 9.02 | 5.52 |
| Gemini 2.0 | 58.6 | 53.7 | 76.1 | 86.9 | 10.12 | 4.91 |
| Story-Adapter | **70.0** | 62.6 | 38.8 | 82.0 | **10.36** | 5.81 |
| Ours | 64.6 | 69.4 | **92.2** | **93.4** | 10.16 | **5.88** |

As shown in Table 6, our method performs strongly **across core dimensions**, particularly in **Alignment**, **OCCM**, and **Aesthetics**. While some baselines achieve slightly higher scores on low-level quality metrics, **LogiStory excels in narrative coherence and visual storytelling**, which aligns with the central goals of our framework. These results further **demonstrate its generalizability beyond LogicTale**.

### B.5 USER STUDY ANALYSIS

We evaluate inter-rater agreement using **Krippendorff's Alpha ↑**, a robust statistical measure for ordinal-scale annotations. Across **30 annotators** and **8 methods**, the agreement scores are: **0.78** for **Instance Consistency**, **0.82** for **Narrative Causality**, **0.70** for **Story Readability**, **0.60** for **Aesthetic Appeal**.

The relatively lower agreement on Aesthetic Appeal is likely due to the **visual similarity** of outputs from methods such as *ConsiStory* and *StoryDiffusion*. To further confirm annotation reliability, we conducted a subset analysis on four representative methods: **SDXL**, **StoryDiffusion**, **Gemini 2.0 Flash**, and **LogiStory**. In this focused comparison, Krippendorff's Alpha improves to: **0.90** for **Instance Consistency**, **0.94** for **Narrative Causality**, **0.84** for **Story Readability**, **0.86** for **Aesthetic Appeal**.

These results demonstrate **high consistency and reliability** of the human evaluation process, particularly when method diversity increases.

## C LOGICTALE BENCHMARK DETAILS AND ANALYSIS

### C.1 LOGICTALE DATASET CONSTRUCTION

To facilitate the development and evaluation of logic-aware story visualization models, we introduce the **LogicTale** dataset. This dataset is specifically designed to assess narrative coherence, causal interpretability, and visual quality. The construction process is as follows:

### C.1.1 STORY COLLECTION AND COMPOSITION

We curated a total of 60 stories, balancing between classical well-known narratives and newly created original stories in a ratio of 3:2. This ensures both diversity and challenge:

- **Classical stories**: Selected from widely known fables, fairy tales, and folklore to provide recognizable plot structures.
- **Original stories**: Authored by professional writers and illustrators, designed to introduce novel situations, abstract concepts, and creative settings that challenge visual logic modeling.

### C.1.2 ANNOTATION PROCESS

Each story in LogicTale is meticulously annotated with the following elements:

- **Story Title and Source**: Including attribution for classical or original content.
- **Full Narrative Text**: Cleaned and standardized for consistency.
- **Character List**: Detailed descriptions of key characters, their appearances, and roles.
- **Visual Logic Chains**: Structured annotation of critical events using triplets in the format `(action, result, importance weight)`, capturing causal dependencies and expected state changes.
- **Difficulty Tag**: Each story is labeled as *Easy*, *Medium*, or *Hard*, based on:
  - *Easy*: 1-2 characters, straightforward interactions, simple causal links.
  - *Medium*: More than 2 characters, plot twists, moderately complex interactions.
  - *Hard*: Multiple characters, abstract events, non-linear storytelling, or temporal jumps.

```
1  {
2      "id": 1,
3      "level": "easy",
4      "title": "The Crow and the Pitcher",
5      "source": "Aesop's Fables",
6      "story_outline": "Under the scorching sun, across a vast, parched land...",
7      "character_list": ["crow"],
8      "causal_event_chain": [
9          {
10             "action": "Crow tries to drink water but fails",
11             "result": "Crow looks for a solution",
12             "weight": 0.3
13         },
14         {
15             "action": "Crow picks up pebbles and drops them into the bottle",
16             "result": "Water level rises",
17             "weight": 0.5
18         },
19         {
20             "action": "Water level reaches the top",
21             "result": "Crow drinks the water",
22             "weight": 0.2
23         }
24     ]
25  }
```

### C.1.3 QUALITY CONTROL

All annotations were performed by experienced annotators with backgrounds in storytelling, visual arts, and education. A multi-phase validation process was conducted to ensure:

- **Annotation Accuracy**: Cross-checking by multiple annotators.
- **Logical Soundness**: Ensuring causal chains are coherent and interpretable.
- **Visual Plausibility**: Verifying that the story could be visually realized in a multi-image sequence.

### C.1.4   DATASET PURPOSE

LogicTale is intended as both a **training resource** for enhancing logic-aware story visualization models and a **benchmark** for evaluating models in terms of visual logic, consistency, and storytelling quality.

### C.2   EVALUATION DETAILS

### C.2.1   AUTOMATIC EVALUATION

We design a comprehensive automatic evaluation framework that covers both **visual logic consistency** and **visual aesthetics**. Specifically, we adopt the following six metrics:

**1. Instance Consistency:**   We employ a Large Language Model (LLM) to assess the consistency of key elements across the image sequence. We designed the following prompt to guide the LLM in evaluating the consistency of key elements (characters, objects, scenes) across the image sequence:

> You are given a story and a sequence of images representing different moments from the story. Your task is to evaluate whether the same characters, key objects, and environments appear consistently and coherently throughout the images.
> Please assess the overall instance consistency using the following scale:
>
> - **1 - Poor:** Severe inconsistencies; characters, objects, or environments change drastically without narrative justification.
> - **2 - Fair:** Multiple inconsistencies present; noticeable attribute or appearance shifts that harm understanding.
> - **3 - Good:** Minor inconsistencies; small differences in appearance or object details, but the overall coherence is mostly maintained.
> - **4 - Very Good:** Mostly consistent with only subtle or hard-to-notice differences.
> - **5 - Excellent:** Fully consistent; characters, objects, and environments are visually stable and coherent across all frames.
>
> Please provide the rating and a brief justification.

**2. Narrative Causality:**   To evaluate key event causality, we adopt a Visual Question Answering (VQA)-based strategy. For each annotated key event in the dataset, we formulate specific questions that probe the causality (e.g., "*Does the wolf fall into the pot after climbing the chimney?*"). The answers generated by the model are compared to the ground truth, and the score is aggregated as the average accuracy over all key causal questions.

**3. Story Readability:**   We adopt a two-step approach. First, an image captioning model (e.g., BLIP-2) generates a textual description for the entire image sequence. Then, the LLM is provided with the story's character list and the generated captions and is tasked to infer the overall story plot. The inferred story is compared to the original story text using text similarity scores computed via CLIP-based embedding similarity.

**4. Aesthetic Score:**   We utilize HPSv2 to automatically evaluate the aesthetic quality of each generated image. The overall score is obtained by averaging the per-image scores across the sequence.

**5. Style Consistency:**   We adopt DINOv2 to compute the visual embeddings of all images in a sequence and measure the pairwise cosine similarity. The higher the average similarity, the more stylistically consistent the image sequence is considered.

**6. Character Expressiveness:**   An LLM is tasked to rate the appropriateness of character expressions and actions with respect to the narrative. The model observes the image sequence and is instructed to assign a score from 1 (poor) to 5 (excellent) based on how well the characters' emotions, gestures, and poses align with the story's development. We designed the following prompt to guide the LLM in evaluating the expressiveness of characters across the image sequence:

You are given a story and a sequence of images representing different moments from the story. Your task is to evaluate whether the characters' emotions, gestures, and body language are clearly conveyed and appropriate to the narrative context. Please assess the overall character expressiveness using the following scale:

- **1 - Poor:** Characters appear expressionless or with irrelevant/unintelligible expressions; emotional intent is entirely unclear.
- **2 - Fair:** Characters show limited or inconsistent expressions; emotions are weakly conveyed and often mismatched with the story context.
- **3 - Good:** Characters display some relevant expressions or gestures, but emotional clarity is only partially achieved.
- **4 - Very Good:** Characters are generally expressive and aligned with the narrative; only minor ambiguities remain.
- **5 - Excellent:** Characters are highly expressive; emotions, gestures, and body language are vivid, coherent, and fully aligned with the story context.

Please provide the rating and a brief justification.

### C.2.2 HUMAN EVALUATION PROTOCOL

To complement automatic evaluation, we conducted a structured human study across four evaluation dimensions. Thirty annotators participated, each being shown the full story text and the corresponding generated image sequences from eight different methods. All tasks were randomized to avoid ordering bias. Below, we detail the protocol for each dimension.

**Instance Consistency (1–5).** Measures whether the same character or entity maintains consistent appearance across the story sequence. Annotators were instructed with the following prompt:

Across the entire image sequence, are the main characters or entities visually consistent in terms of identity, clothing, and major attributes? Please provide a score from 1 to 5, following the scale below, and a brief justification:

- **1 - Poor:** Severe inconsistencies; characters, objects, or environments change drastically without narrative justification.
- **2 - Fair:** Multiple inconsistencies present; noticeable attribute or appearance shifts that harm understanding.
- **3 - Good:** Minor inconsistencies; small differences in appearance or object details, but overall coherence is mostly maintained.
- **4 - Very Good:** Mostly consistent with only subtle or hard-to-notice differences.
- **5 - Excellent:** Fully consistent; characters, objects, and environments remain visually stable and coherent across all frames.

**Narrative Causality (binary judgment).** Since LogicTale provides ground-truth causal chains, we designed a VQA-style evaluation where annotators were asked causal questions derived from dataset annotations. For each causal pair (*action → result*), we generated prompts such as:

In this story, after [action], does [result] happen in the images? (Yes/No)

Scores were aggregated as the percentage of correctly satisfied causal relations.

**Story Readability (1–5).** Reflects how well the story can be understood solely through the image sequence. Annotators were instructed with the following prompt:

If you only look at the images without re-reading the text, how clearly can you understand the intended narrative progression? Please provide a score from 1 to 5, following the scale below, and a brief justification:

- **1 - Poor:** Completely confusing; the story is impossible to follow.
- **2 - Fair:** Very unclear; only fragments of the story are understandable.
- **3 - Good:** Partially clear; the overall idea is somewhat recognizable but many gaps remain.

- **4 - Very Good:** Mostly clear; only minor ambiguities remain.
- **5 - Excellent:** Very clear; the story is easy to follow without text.

**Aesthetic Appeal (1–5).** Captures perceptual quality, including realism, visual attractiveness, and stylistic coherence. Annotators were instructed with the following prompt:

> Considering the image sequence as a whole, how visually appealing and stylistically coherent are the generated images? Please provide a score from 1 to 5, following the scale below, and a brief justification:
>
> - **1 - Poor:** Very low quality; unrealistic and inconsistent images.
> - **2 - Fair:** Low quality; distracting artifacts or mismatched styles.
> - **3 - Good:** Moderate quality; acceptable but with noticeable flaws.
> - **4 - Very Good:** High quality; visually appealing with only minor issues.
> - **5 - Excellent:** Excellent quality; highly realistic and aesthetically pleasing.

Each image sequence was independently evaluated by at least 5 annotators, and the final score was computed by averaging the ratings.

### C.3 ABLATION STUDY ON LOGICTALE: DATASET SCALE AND EVALUATION STABILITY

To evaluate whether a relatively small but high-quality dataset like LogicTale can yield stable and discriminative evaluation, we conduct a saturation analysis across subsets of increasing size. Specifically, we construct incremental subsets $\mathcal{D}_{12} \subset \mathcal{D}_{24} \subset \cdots \subset \mathcal{D}_{60}$, maintaining a balanced distribution of story difficulty and topic. For each subset, we evaluate all competing models using key automated metrics such as visual logic coherence and story understanding similarity.

We analyze the consistency of model rankings, measured using Kendall's Tau correlation against the full 60-story set. Figure 12 shows that the model rankings stabilize as the subset size increases. This demonstrates that even with 60 carefully curated stories, LogicTale provides a reliable and discriminative benchmark for visual logic assessment.

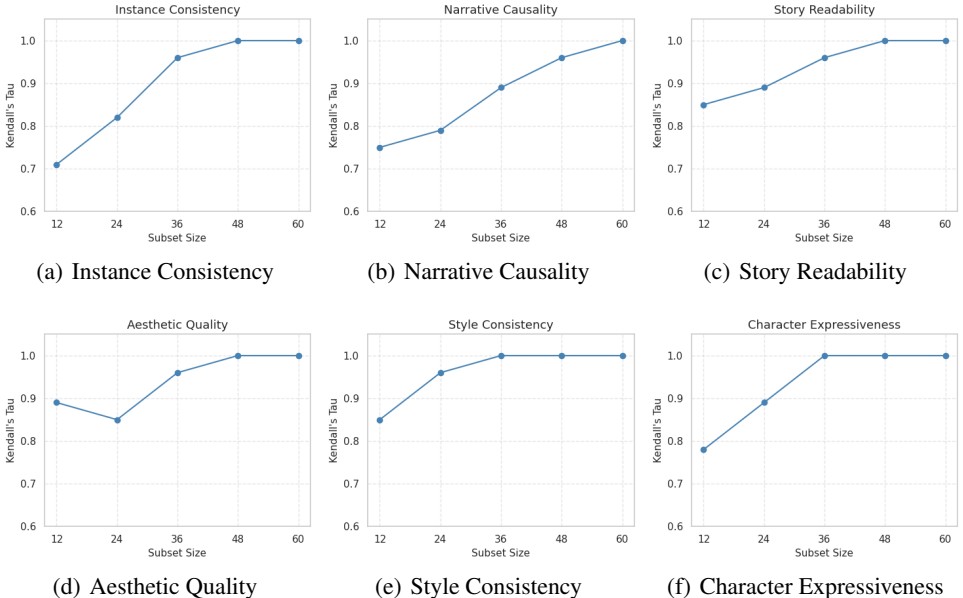

Figure 12: Kendall's Tau correlation between model rankings on full dataset (60 stories) and various dataset subsets across six evaluation metrics. Results indicate high ranking consistency with subsets as small as 36 stories, validating the reliability of LogicTale for evaluation purposes.

| Dataset | #Samples | Logic Annotation | Evaluation Mode |
|---|---|---|---|
| StoryDiffusion | 20 | ✗ | Single-image |
| ConsiStory | 20 | ✗ | Single-image |
| MM-StoryAgent | 100 | ✗ | Single-image |
| MovieAgent | 12 | ✗ | Single-image |
| ViStoryBench | 80 | ✗ | Single-image |
| **LogicTale (Ours)** | 60 | ✓ | **Multi-image joint** |

Table 7: Comparison of **LogicTale** with existing story visualization benchmarks. Unlike prior datasets, **LogicTale** provides explicit causal logic annotations and supports **multi-image joint evaluation**, enabling systematic assessment of narrative coherence.

## C.4 DATASET COMPARISON

As shown in Table 7, existing datasets for story visualization are limited in both scale and scope: most consist of short, single-character or handcrafted narratives, lack logical annotations, and restrict evaluation to single-image fidelity or alignment. In contrast, **LogicTale** is the **first dataset** that explicitly incorporates **causal logic annotations**, enabling systematic evaluation of story-level reasoning. Furthermore, we design a suite of **automatic multi-image evaluation metrics** that go beyond local image quality to assess **cross-image semantic consistency and narrative causality**, providing a more rigorous foundation for advancing logic-aware story visualization.

## D LOGICSTORY FRAMEWORK DETAILS

### D.1 LOGIC-AWARE MULTI-AGENT SYSTEM

To generate structured and logic-aware visual narratives, **LogiStory** employs a multi-agent planning system composed of three agents: *SceneCrafter*, *LogicMiner*, and *ShotPlanner*. These agents work in sequence to analyze the story text and construct an intermediate story plan, consisting of characters, key events, and frame-level scene descriptions.

### D.1.1 SCENECRAFTER

The *SceneCrafter* agent is responsible for extracting the visually relevant components of the story, including characters, key objects, and environmental settings. Unlike conventional entity extraction, we ask the model to infer and provide visual attributes such as appearance, size, texture, and emotional tone. This provides strong guidance for subsequent visual generation.

**Prompt Example:**

> You are a visual designer tasked with preparing for story illustration. Given a short story, identify and describe the following elements with visual details:
>
> - **Characters:** List each character. For each, provide a name and a short description of their physical appearance, clothing, emotional traits, and any distinguishing features.
> - **Key Objects:** Identify important objects mentioned or implied in the story. Describe their size, material, and visual appearance (e.g., color, texture).
> - **Scene Locations:** Identify each main scene or background. Describe the environment, including style (e.g., cartoon, watercolor), atmosphere (e.g., cozy, ominous), and key visual elements (e.g., forest path, house made of straw).
>
> **Your output should be structured in three sections: Characters / Key Objects / Scene Locations. Be as visual as possible.**
>
> Story:    "The three little pigs leave their mother to build their own houses.  The first pig buys straw,

```
the second buys wood, and the third buys bricks.   Each
builds a house from their chosen material."
```

**Expected Output Format:**

- **Characters:**
    - **Pig1:** A small, cheerful piglet wearing a blue shirt. Has round eyes and a playful smile. Carries a bundle of straw.
    - **Pig2:** Medium-sized pig with a green cap and suspenders. Appears cautious and focused. Often seen holding wooden planks.
    - **Pig3:** Slightly larger pig with glasses and a red scarf. Looks serious and thoughtful. Holds bricks in a neat stack.
    - **Mother Pig:** A kind, elderly pig wearing an apron and bonnet. Stands at the cottage door waving goodbye.
    - **Wolf:** A tall, menacing wolf with gray fur, sharp eyes, and a mischievous grin. Often lurking in the background.
- **Key Objects:**
    - **Straw bundle:** Light yellow, loosely tied, lightweight with uneven edges.
    - **Wood planks:** Brown, sturdy, rectangular with visible grain texture.
    - **Brick stack:** Dark red, neatly aligned bricks with mortar stains.
    - **Boiling pot:** Large iron cauldron with steam rising from it, placed over firewood.
- **Scene Locations:**
    - **Forest path:** Gentle trail surrounded by tall green trees, soft lighting, storybook cartoon style.
    - **Pig1's straw house:** Small, straw-covered hut with slanted roof. Cozy but fragile appearance.
    - **Pig2's wood house:** Medium-sized cabin with wooden logs and a chimney. Structured and rustic.
    - **Pig3's brick house:** Solid and square, with tiled roof and brick walls. Brightly lit and secure.
    - **Mother's home:** A warm cottage with a garden, classic watercolor storybook feel.

This visually grounded output equips the system with rich contextual signals, serving as a scaffold for logic mining and shot planning in downstream modules.

### D.1.2 LogicMiner

The *LogicMiner* agent extracts a sequence of causally and temporally significant events from the story text. Each event is represented using a structured 4-tuple format **(actor, action, target, result)**, capturing both physical interactions and narrative consequences. This formalization ensures fine-grained reasoning and facilitates downstream consistency checking and scene planning.

**Prompt Example:**

You are a story logic analyst. Given a short narrative, your task is to extract the key events that define the causal and temporal structure of the story. For each event, output a structured record in the format:

**(actor, action, target, result)**

Where:

- **actor** is the entity initiating the event.
- **action** is the main verb or activity.
- **target** is the object or entity affected by the action.
- **result** describes what changed in the world due to this event.

Your output should be a chronological list of events. After each event, also specify:

- **Preconditions:** What must be true before the event occurs.
- **Effects:** What changes after the event (state change).

Story: "The first little pig buys straw and builds a house. The wolf comes and blows it down. The pig runs to the second pig's wooden house. The wolf follows and destroys that house too. Both pigs escape to their brother's brick house. The wolf tries to enter through the chimney but falls into a pot of boiling water."

**Expected Output Format:**

1. (Pig1, buys, straw, Pig1 owns building material)
   *Preconditions: Straw is available for purchase.*
   *Effects: Pig1 has straw.*

2. (Pig1, builds, house, Straw house exists)
   *Preconditions: Pig1 owns straw.*
   *Effects: Straw house appears in the scene.*

3. (Wolf, blows down, straw house, House destroyed, Pig1 is homeless)
   *Preconditions: Straw house exists. Wolf is nearby.*
   *Effects: Straw house is destroyed. Pig1 must flee.*

4. (Pig1, runs to, Pig2's house, Pig1 and Pig2 are together)
   *Preconditions: Pig1 is outside. Pig2's house is standing.*
   *Effects: Pig1 is inside Pig2's house.*

5. (Wolf, blows down, wood house, House destroyed, Pig1 and Pig2 flee)
   *Preconditions: Wood house exists. Wolf is nearby.*
   *Effects: House destroyed. Two pigs run away.*

6. (Pig1 and Pig2, escape to, Pig3's house, All pigs are in brick house)
   *Preconditions: Pig3's house is intact.*
   *Effects: All three pigs are together in brick house.*

7. (Wolf, climbs, chimney, Wolf attempts to enter house)
   *Preconditions: All doors and windows are shut.*
   *Effects: Wolf is inside the chimney.*

8. (Pigs, boil, water, Boiling water is prepared)
   *Preconditions: Pot and fire are present.*
   *Effects: Trap for the wolf is ready.*

9. (Wolf, falls into, boiling pot, Wolf is defeated)
   *Preconditions: Wolf is in chimney. Pot is boiling.*
   *Effects: Wolf is burned and flees. Pigs are safe.*

This structured output enables later modules to verify logical alignment across frames and ensures causal coherence in the resulting visual sequence.

### D.1.3 SHOTPLANNER

The *ShotPlanner* agent bridges structured event logic and visual storytelling. Given the original story text and the event list extracted by the *LogicMiner*, it generates a sequence of shot plans. Each shot plan specifies the visual composition for an image, including **characters, actions, objects, spatial relations, scene context**, and **camera parameters** (e.g., angle, shot type, distance). The planner also produces a rendering prompt in *Stable Diffusion* format for each image.

To enhance narrative clarity and visual engagement, ShotPlanner incorporates principles of **visual storytelling conventions**, such as:

- **Pacing:** Allocate longer visual emphasis to high-impact events (e.g., conflict, climax).
- **Framing:** Use wide, medium, or close-up shots to vary focus and emotional tone.

- **Perspective:** Adjust camera angle and viewpoint to highlight relationships, danger, or tension.

**Input:**

- Story text: A paragraph-length narrative.
- LogicMiner event list: A chronological list of (actor, action, target, result) tuples.

**Output (for each frame):**

- **Shot Plan:**
    - *Scene Description:* Natural language summary of what happens in the shot.
    - *Key Elements:* {characters, actions, objects, spatial layout, background}
    - *Camera Setup:* {shot type (e.g., wide, medium, close), angle (e.g., eye-level, low-angle), focal length}
- **Rendering Prompt (Stable Diffusion format):** A concise visual prompt used for generation.

**Prompt Template:**

You are a visual story director. Given the following story and the structured list of events, your task is to design a sequence of image shots that visually depict each event in a narratively coherent and aesthetically pleasing way.

For each event, provide:

1. **Scene Description:** What is happening in the scene?
2. **Characters and Actions:** Who is doing what?
3. **Objects and Scene Elements:** What objects or environment features are involved?
4. **Spatial Layout:** Where is each character/object located in the scene?
5. **Camera Parameters:**
    - **Shot Type:** (e.g., wide shot, over-the-shoulder, close-up)
    - **Camera Angle:** (e.g., eye-level, high-angle, low-angle)
    - **Perspective:** (e.g., character perspective, bird's eye)
6. **Rendering Prompt (Stable Diffusion Style):** Include:
    - scene setting and characters
    - emotional tone
    - composition and angle
    - style (e.g., children's book illustration, watercolor)

**Story:** "The first little pig builds a straw house. The wolf blows it down. The pig runs away. The second pig builds a wooden house..."

**Events:**

- (Pig1, builds, house, Straw house exists)
- (Wolf, blows down, straw house, House destroyed)

**Example Output:**

**Shot 1:**

- *Scene Description:* Pig1 is happily finishing the construction of a straw house in a sunny field.
- *Characters and Actions:* Pig1 is placing the final straw on the roof.
- *Objects and Scene Elements:* Straw house, piles of straw, a hammer.
- *Spatial Layout:* Pig1 stands in front of the straw house, facing the viewer.

- *Camera Parameters:*
  - Shot Type: Medium shot
  - Camera Angle: Eye-level
  - Perspective: Third-person perspective
- *Rendering Prompt (Stable Diffusion):* "A cheerful pig building a straw house in a sunny meadow, children's book illustration style, medium shot, eye-level angle, bright and warm color palette, watercolor texture, detailed background with green trees and blue sky"

**Shot 2:**

- *Scene Description:* The wolf is blowing down the straw house while Pig1 watches in fear.
- *Characters and Actions:* Wolf is exhaling forcefully; Pig1 is covering its face.
- *Objects and Scene Elements:* Straw flying, partially collapsing house.
- *Spatial Layout:* Wolf on left, Pig1 on right, house between them.
- *Camera Parameters:*
  - Shot Type: Wide shot
  - Camera Angle: Slight low-angle to emphasize action
  - Perspective: Dynamic third-person
- *Rendering Prompt (Stable Diffusion):* "A fierce wolf blowing on a fragile straw house while a scared pig watches, straw flying everywhere, wide shot, low angle, dramatic lighting, children's book illustration, vivid cartoon style"

This module ensures that the resulting image sequence not only aligns with the logical events but also delivers a cinematic and emotionally engaging viewing experience, tightly coupling narrative rhythm with visual coherence.

### D.2 VISUAL LOGIC ENHANCEMENT MODULE

#### D.2.1 LOCAL CAUSAL MONITOR

The *Local Causal Monitor* simulates a human-like reading experience by incrementally assessing the logical consistency of each image in a story sequence, conditioned on prior visual context and world knowledge. Unlike global verification, which evaluates the entire story retrospectively, this module performs **step-by-step causal validation** during sequence unfolding, modeling the linear comprehension process of human readers.

**Core Design:** At each timestep $t$, the monitor:

1. Maintains a **Memory Buffer** $M_{t-1}$ summarizing all previously observed visual content and inferred world states.
2. Parses the current image $I_t$ into a structured caption or event description using an image captioning or event extraction model.
3. Evaluates whether $I_t$ is logically compatible with $M_{t-1}$ using a large language model (LLM).
4. Assigns a **causal consistency score** $s_t \in [0, 1]$ representing the degree of logical alignment.
5. Updates the buffer $M_t$ by incorporating the new information from $I_t$.

This mechanism enables fine-grained monitoring of temporal consistency, causal progression, and state transitions across a story sequence.

**Memory Buffer Example:**

> **Initial Story:** "Three little pigs each build a house using straw, wood, and bricks. A wolf tries to blow each house down."

**Memory after Image 1 (Pig1 builds straw house):** `"Pig1 has constructed a straw house in an open field. No threats have appeared yet. Other pigs are not present."`

**Memory after Image 2 (Wolf blows down straw house):** `"Pig1's straw house has been destroyed by the wolf. Pig1 is frightened and escapes. The wolf is now active in the story."`

**Image 3 (Current):** Pig2 is shown relaxing in a completed wooden house, unaware of any danger.

**Question:** Based on the current memory and this image, is the event causally consistent with the story flow?

**LLM Evaluation Prompt:**

You are a causal reasoning expert.

Given: - The memory of previously observed story events - The current image description

Evaluate whether the current image is **logically consistent** with prior context, considering: 1. Whether the sequence of events makes causal sense 2. Whether character behavior is appropriate given past events 3. Whether any contradictions or unexplained jumps occur

Return a numerical consistency score between 0 (completely inconsistent) and 1 (fully consistent), along with a brief justification.

**Memory Buffer:** Pig1's straw house was destroyed by the wolf. Pig1 ran away. The wolf is now active in the story.

**Current Image Description:** Pig2 is relaxing in a newly built wooden house, smiling. No signs of the wolf or alarm.

**Output Format:** Score: <float between 0 and 1> Justification: <one or two sentences>

**Example Answer:** Score: 0.8 Justification: The scene is mostly logical. Pig2 may not yet be aware of the wolf's actions, which explains the relaxed demeanor.

**Scoring Interpretation:**

- **Score = 1.0:** Full causal alignment with prior context
- **Score ∈ [0.7, 0.9]:** Minor temporal gaps or ambiguity, still plausible
- **Score ∈ [0.4, 0.6]:** Noticeable logical inconsistencies, partially recoverable
- **Score < 0.4:** Major contradiction or missing transitions

This local monitor provides frame-level causal validation and supports training or evaluation by detecting inconsistencies early during visual narrative generation.

**Threshold Calibration for Image Refinement.** To determine the appropriate response for each generated panel $p_t$ during story visualization, we introduce two thresholds, $\tau_1$ and $\tau_2$, which guide the image refinement process based on the normalized logic consistency score $\psi(p_t \mid \mathcal{M}_{t-1})$. This score integrates outputs from both the Local Causal Monitor and the Global Causal Verifier, representing the confidence that $p_t$ aligns with the intended narrative logic.

We define:

- $\tau_1$: the upper bound below which the image is considered **logically invalid** and must be **regenerated**.
- $\tau_2$: the lower bound above which the image is considered **logically acceptable**, requiring **no refinement**.
- $[\tau_1, \tau_2]$: a range indicating **partial alignment**, where the image is passed through a refinement stage using inpainting or MLLM-based editing.

To empirically determine these thresholds, we conducted a calibration study on a held-out validation set:

1. A diverse set of generated panels was sampled from stories of varying complexity.
2. Human annotators rated the logical alignment of each panel with the narrative on a 0–5 Likert scale.
3. The corresponding logic scores $\psi$ were collected from our verifier modules.
4. A distributional analysis was performed to align human judgments with $\psi$.

As a result:

- $\tau_1$ was set to **0.4**, covering the 90th percentile of panels rated below 2 (logically flawed).
- $\tau_2$ was set to **0.7**, corresponding to the 10th percentile of panels rated above 4 (logically sound).

This calibration ensures that the refinement mechanism is grounded in human-perceived narrative coherence, aligning automated validation with qualitative standards.

### D.2.2 GLOBAL CAUSAL VERIFIER

The *Global Causal Verifier* is designed to ensure **global narrative coherence** throughout the visual story generation process. It constructs and leverages a high-level **Causal Graph** that models the dependencies between key events, character/object states, and their temporal transitions, enabling comprehensive evaluation and refinement across the entire sequence.

**Causal Graph Construction:** We first construct a directed **Global Causal Graph** $G = (V, E)$ from the story text and key events extracted by the `LogicMiner`. Each node $v_i \in V$ represents a distinct story state or event, and each edge $e_{ij} \in E$ denotes a causal or temporal dependency such as:

- *Event causality:* "Pig builds house" $\rightarrow$ "Wolf tries to blow it down"
- *State evolution:* "Pot placed under chimney" $\rightarrow$ "Wolf falls into pot"
- *Implicit physics:* "Stones added to pot" $\rightarrow$ "Water level rises"

Each node is annotated with involved characters, object states, spatial relations, and expected visual outcomes.

**State Recorder:** During generation, we maintain a **State Recorder** $S_t$ that tracks which portion of the global story has been visually realized up to timestep $t$. It summarizes observed character locations, object configurations, known outcomes, and remaining events.

For example:

> At $t = 3$ (third image): `"Pig1's house destroyed, Pig1 escaped, Pig2 building house, wolf seen approaching"`

This enables the verifier to identify mismatches or missing transitions between the current visual state and expected causal flow.

**Refinement Instruction Generation:** At each step, the current image $I_t$ is parsed into a structured scene description (e.g., via captioning or scene graph parsing). The verifier then checks for:

- **Causal Alignment:** Does $I_t$ align with the next expected node in $G$?
- **State Progression:** Are object/character states consistent with prior evolution?
- **Implicit Logic:** Does $I_t$ reflect necessary physical/visual reasoning (e.g., gravity, containment)?

If misalignment is detected, a **refinement instruction** is generated using a language model, aimed at correcting the image in the next iteration.

**Refinement Prompt (LLM):**

You are a visual reasoning assistant.

Given: - The global causal graph for the story - The current generation state ($S_t$) - The current image description - The next expected story event

Determine whether the current image faithfully represents the expected event. If not, generate a concise instruction for image refinement, focusing on correcting logic or state alignment.

**Example:**

- **Expected Event:** "Wolf climbs chimney, pigs prepare boiling pot"
- **Current Image Description:** "Wolf near house, pigs inside, pot missing"
- **Instruction:** "Add boiling pot beneath chimney. Show pigs anxiously watching chimney."

This refinement instruction is passed to the visual generation module for image editing or regeneration, ensuring both visual fidelity and narrative coherence. By integrating top-down symbolic planning with bottom-up visual verification, the Global Causal Verifier enforces long-range consistency and supports correction of subtle causal gaps that may not be captured at the local level.

## E    EXPERIMENT COMPUTE RESOURCES AND SETTINGS

All experiments were conducted on a workstation equipped with two NVIDIA A6000 GPUs (48GB VRAM each). All baseline models, including StoryDiffusion and ConsiStory, were used with their officially released checkpoints and settings for fair comparison. No additional fine-tuning was performed on these models unless specified.

## F    DISCUSSION

### F.1    ADVANTAGES

Our proposed framework **LogiStory** and the curated dataset **LogicTale** offer multiple advantages to the field of visual storytelling and beyond:

**1. Explicit Modeling of Visual Logic.**    Unlike prior approaches that implicitly rely on latent representations, LogiStory explicitly models *visual logic*—defined as the perceptual and causal coherence among characters, actions, and object states—through a structured multi-agent planning mechanism and logic-aware verification modules. This improves not only the factual correctness of individual images but also their temporal and narrative coherence.

**2. Fine-Grained Story Understanding.**    Through the `SceneCrafter`, `LogicMiner`, and `ShotPlanner` agents, our system decomposes stories into interpretable components (e.g., characters, key events, spatial relations, and camera instructions), enabling transparent, controllable, and explainable generation. This opens the door for deeper interaction and analysis in multimodal generation.

**3. Logic-Aware Image Refinement.**    By integrating both **Local Causal Monitor** and **Global Causal Verifier**, LogiStory simulates human-like reading and comprehension processes, identifying inconsistencies both at the fine-grained image level and the story-wide causal structure. This dual-level feedback loop guides generation toward visually and narratively coherent outcomes.

**4. LogicTale Dataset as a Structured Benchmark.**    LogicTale provides rich annotations including causal graphs, key events, object states, and visual prompts, serving as a high-quality benchmark for evaluating visual logic in story generation. It supports multiple evaluation paradigms—automatic metrics, human judgment, and logic tracing—and can foster the development of explainable multimodal models.

**5. Generalizability Beyond Story Visualization.** While our work focuses on the multi-image story visualization task, the underlying principles of visual logic modeling and logic-aware planning generalize naturally to related areas such as:

- **Video Generation:** Ensuring temporal consistency and causal flow in generated frames.
- **Interactive Narrative Systems:** Providing feedback for story-based games or simulations.
- **Multimodal Planning and Robotics:** Translating language instructions into logically consistent visual action sequences.

This makes LogiStory a versatile foundation for broader multimodal reasoning and generation tasks.

### F.2 LIMITATIONS

Despite its strengths, our approach presents several limitations that highlight future research directions:

**1. Scalability to High-Entity Scenarios.** When the story involves a large number of characters, objects, and complex interdependencies, maintaining consistent identity, state, and spatial logic across frames becomes increasingly difficult. This reflects a broader limitation in current generative models, **even state-of-the-art models such as GPT-4o struggle to guarantee visual identity preservation and detailed multi-object reasoning during image editing or generation**.

**2. Framework Complexity.** Our framework adopts a modular multi-agent design that allows flexibility, but it also introduces additional computational steps compared to simpler pipelines. These trade-offs reflect our emphasis on interpretability and logical rigor, and we believe they open avenues for future refinement rather than fundamental limitations of the approach.

## G USAGE OF LLMS

Large language models (LLMs) were used solely as an aid to polish the language and improve the clarity of exposition. They were **not involved in research ideation, problem formulation, methodology design, experimental implementation, or analysis of results**. All scientific contributions and claims are the work of the authors.

