# OpenReview forum: "LogiStory: A Logic-Aware Framework for Multi-Image Story Visualization"
_ICLR.cc/2026/Conference — ICLR 2026 Poster_

### Official Review · Reviewer_wuEE · 2025-10-28

**Soundness:** 3
**Presentation:** 4
**Contribution:** 3
**Rating:** 8
**Confidence:** 3

**Summary:**

This paper proposes a logicaware multi-image story visualization framework, LogiStory. The framework is built around the central innovation of explicitly modeling visual logic in story visualization. This paper also introduces LogicTale, a benchmark comprising richly annotated stories, emphasizing causal reasoning, and visual logic interpretability.

**Strengths:**

- This paper introduces LogicTale, a new benchmark comprising richly annotated stories based on not only classical stories but also original stories, to validate on new stories.
- This paper also proposes a logicaware multi-image story visualization framework, LogiStory to address the challenges of visual consistency and logical coherence in a series of story visualizations by constructing a new benchmark dataset. The results show high performance of LogiStroy on LogicTale.
- LogiStroy was evaluated on ViStoryBench-Lite to assess the generalizability of the framework. Ablation studies are also included to evaluate each module.

**Weaknesses:**

- The framework is composed of a complex pipeline based on a multi-agent system.
- The evaluation dimensions using LogiStory and the focus points of the agents are closely related, which raises the possibility that the framework may be somewhat dependent on the evaluation aspects.

**Questions:**

- It was stated that "no existing benchmark systematically measures whether a visual sequence, either image-based or video-based, successfully communicates the intended story in a logically coherent manner," but VinaBench[1] is a benchmark dataset that considers commonsense links and visual consistency as visual narrative generation.

[1] VinaBench: Benchmark for Faithful and Consistent Visual Narratives（CVPR2025）

- Will the benchmark dataset, source code, and checkpoints be made publicly available?

---

> ### Author Response · Authors · 2025-11-19
> **Rebuttal (1/2)**
>
> We sincerely thank the reviewer for the insightful feedback. Below we respond to each concern raised:
>
> **W1: System Design**
>
> Below we clarify the motivation and demonstrate why the design does not hinder generality or extensibility.
>
> - **Design Motivation: Human-Like Decomposition for Story Visualization**
>
>   The multi-agent formulation is intentionally inspired by how **human creators naturally decompose story visualization tasks**. When producing a storyboard or directing a short film, creators rarely operate as a single “all-in-one” unit. Instead, they divide responsibilities into
>
>   - Establishing characters and environments
>
>   - Analyzing causality and implicit events
>
>   - Planning shots and camera movements
>
>   This decomposition is essential for achieving **coherence, interpretability, and control**. Our agents mirror this practice by assigning distinct and modular roles to SceneCrafter, LogicMiner, and ShotPlanner. This structure allows the system to handle the inherently multi-faceted nature of story visualization more effectively than a monolithic planner.
>
> - **Engineering Complexity Does Not Imply Limited Generality**
>
>   The multi-agent pipeline is not gratuitous engineering; rather, it provides **structure that improves generalization and logical coherence**. Our ablation studies confirm this:
>
>   - **Replacing the multi-agent planner with a single LLM-based planner** leads to clear performance degradation, especially in narrative causality and readability. This indicates that the decomposition is functionally beneficial rather than an arbitrary engineering choice.
>
>   - **Evaluations across different model backbones** show that the framework maintains its advantages even when using smaller or alternative LLMs. This demonstrates that the approach is not tied to a specific model size or vendor, supporting its adaptability.
>
>   Together, these results show that **engineering complexity in our framework does not equate to limited methodological applicability**. Instead, the modularity ensures interpretability, plug-and-play flexibility, and robustness across settings.
>
>
>
> **W2: Potential Coupling Between Evaluation Dimensions and Agent Design**
>
> Below we clarify why the evaluation protocol does not bias the framework, and why the improvements are **genuine** rather than metric-specific.
>
> - **Our work *intentionally* targets story logic and multi-image interpretability**
>
>   The core goal of LogiStory is to address **narrative coherence**, **causal clarity**, and **multi-image story interpretability**, dimensions largely underexplored by existing frameworks.
>   Because our method explicitly aims to improve **story-level logic**, it is natural that our benchmark measures these properties. However, this does not imply overfitting. All proposed metrics are grounded in standard narrative theory and causal reasoning, and our model’s internal agents focus on these aspects because **they are the central research problem addressed by the work**, not because of alignment with any particular score.
>
> - **Multiple external and heterogeneous dimensions evaluation**
>
>   Our experiments include broad and diverse evaluations that go well beyond the LogicTale metrics introduced in our work.
>   Specifically:
>
>   - We evaluate on **VistoryBench-Lite**, an external benchmark with its own narrative-consistency criteria and scoring protocol.
>
>   - We additionally report standard **visual quality and representation metrics** such as Aesthetic Score, DINOv2 embedding similarity, and CLIP-based measures. These external metrics are not optimized by any agent in our pipeline.
>
>   Since LogiStory consistently outperforms baselines on these dimensions, the empirical results indicate that our improvements **generalize beyond our own proposed evaluation scheme**.
>
> - **User study results verify improvements beyond automated metrics**
>
>   Most importantly, human evaluation provides direct evidence that the gains are not tied to any specific metric.
>   In user study, participants preferred LogiStory over all baselines in terms of:
>
>   - **narrative clarity**,
>
>   - **causal interpretability**
>
>   - **overall story readability**.
>
>   Human preference is inherently **metric-agnostic** and cannot be overfitted through agent design. The fact that LogiStory achieves the highest human scores strongly supports that the improvements arise from genuinely better story reasoning and visualization, rather than from tuning toward specific evaluation dimensions.
>
> Taken together:
>
> - our work is fundamentally focused on story logic and interpretability by design,
> - the framework performs strongly on external benchmarks and heterogeneous metrics, and
> - human studies confirm genuine qualitative improvements.
>
> These results collectively demonstrate that LogiStory does **not** depend on any single evaluation dimension, and the gains reflect **true enhancements in multi-image narrative coherence and causal understanding**.

---

> ### Author Response · Authors · 2025-11-19
> **Rebuttal (2/2)**
>
> **Q1: Regarding VinaBench**
>
> We appreciate the reviewer for pointing out the relevance of **VinaBench (CVPR 2025)**. Our original statement was overly absolute, and we have **updated the description in the revised PDF** to more accurately reflect prior work on visual narrative evaluation.
>
> - **Complementarity Between VinaBench and LogicTale**
>
>   Both VinaBench and LogicTale aim to assess narrative coherence, but their **focuses are fundamentally different yet complementary**.
>
>   - VinaBench primarily evaluates **commonsense links** and **visual consistency** at the narrative level, focusing on plausibility between events and overall intra-sequence consistency.
>   - LogicTale provides **explicit structured causal annotations** (event triplets with weighted causal relations) and **panel/shot-level story scripts**, which enable fine-grained **causal verification during generation** through our Local and Global Causal modules.
>   - LogicTale further introduces a **story interpretability metric**, where a model-generated sequence is used to reconstruct the full story via an LLM, allowing automatic assessment of whether the intended narrative can be faithfully communicated.
>
>   These distinctions make the two benchmarks complementary: VinaBench stresses **commonsense plausibility**, while LogicTale emphasizes **structured causality and story reconstructability**, which are required for our logic-aware generation framework.
>
>
>
>
> **Q2: Code&Dataset Release**
>
> We appreciate the reviewer’s attention to reproducibility.
> To clarify, all resources associated with **LogiStory** will be made publicly available upon acceptance.
>
> **Planned Releases**
>
> - **Complete LogicTale dataset** with annotations, detailed documentation.
> - **Full framework implementation**, including agent prompts, causal verification modules, and generation pipeline scripts (Already in the appendix in text form).
>
> All components will be carefully organized, cleaned, and annotated before release to ensure that the community can easily reproduce, extend, and evaluate our framework.

---

### Official Review · Reviewer_dd8Q · 2025-10-30

**Soundness:** 3
**Presentation:** 3
**Contribution:** 2
**Rating:** 4
**Confidence:** 3

**Summary:**

This paper addresses a significant and well-known challenge in visual sequence generation: the lack of logical and causal coherence. The authors argue that while current models excel at generating high-fidelity images, the resulting sequences often fail as narratives, presenting disjointed actions and fragmented storylines.

**Strengths:**

- The problem proposed in this paper is convincing. The motivation of the paper is very clear. It correctly identifies a key gap in generative models—moving beyond single-image fidelity to narrative coherence. The explicit modeling of "visual logic" is a significant conceptual contribution.

- The creation of the LogicTale benchmark is a valuable contribution to the community.

**Weaknesses:**

- The proposed framework is complex, involving three distinct planner agents, a global causal graph, a state recorder, a local memory buffer, and a two-stage verification loop.

- The scale of the benchmark is too small to fully evaluate the method.

**Questions:**

- I am curious about the computational cost required to generate a story. How many refinement steps (either full regeneration or editing) are typically required per story? Does this number vary significantly with the 'Hard' difficulty stories?

- How robust is this LogicMiner agent to variations in narrative style? Does it perform equally well on highly implicit causality (e.g., "The dog, remembering his master, felt sad") versus the explicit physical causality shown in the examples (e.g., "crow put pebbles into the cup," -> "rising water level" )?

- The baseline naming in the results is slightly inconsistent. Section 5.1 lists GPT-4-image-1 as a closed-model baseline. However, Table 1 lists GPT4o-+GPT-image-1 , and Section 5.2.1 discusses the performance of GPT-4o. Could you clarify if these are the same model, or if GPT40-+GPT-image-1 implies a pipeline using two different models?

- The qualitative results are very strong. To help the community understand the current boundaries of LogiStory, would it be possible to include a qualitative example of a failure case, perhaps from one of the 'Hard' stories? This would provide valuable context for the acknowledged limitation in "high-entity scenarios".

---

> ### Author Response · Authors · 2025-11-19
> **Rebuttal (1/2)**
>
> We sincerely thank the reviewer for the insightful feedback. Below we respond to each concern raised:
>
> **W1: Framework Complexity**
>
> - **The overall system complexity reflects the intrinsic difficulty of the task rather than unnecessary engineering.**
>   Multi-image story visualization inherently requires *planning*, *shot decomposition*, and *iterative refinement*, steps that human creators also follow when producing coherent narratives. Capturing causal dependencies, maintaining entity continuity, and structuring multi-panel scenes cannot be reliably achieved with a single-stage model, thus necessitating a modular design aligned with the nature of the problem.
> - **Ablation studies (Tables 2 and 3) empirically validate that each module contributes meaningfully to narrative coherence and visual logic.**
>   Removing any component, planner agents, the global causal graph, or the verification loop, leads to clear degradation in Narrative Causality, Story Readability, or Instance Consistency. This demonstrates that the system’s structure is not redundant but functionally necessary for the behaviors it enables.
> - **Although the framework is modular, each component can be replaced with simplified alternatives if desired.**
>   For example, the multi-agent planner can be substituted with a *single-agent script planner*, and the global verifier can be replaced with lightweight heuristics. However, as shown in the ablations, such simplifications **consistently reduce narrative-level performance**, confirming that the proposed modules strike a practical balance between complexity and effectiveness.
>
>
>
>
>
> **W2: Dataset Scale**
>
> - **LogicTale is intentionally designed as a high-density, high-cost causal annotation dataset**, where each story requires structured event chains, weighted causal tuples, and character-level definitions. Given the emphasis on narrative causality, **annotation quality is substantially more important than raw dataset size**.
> - **The overall scale of LogicTale is comparable to, or larger than, existing visual-story benchmarks**, which typically contain *20–80 stories* (e.g., StoryDiffusion, ConsiStory, MovieAgent, VisStoryBench). Thus, the dataset is consistent with community practice for causality-focused evaluation.
> - **To further verify generalization, we additionally evaluate LogiStory on ViStoryBench-lite**, which contains out-of-distribution stories. The results (Table 6) demonstrate that our method maintains strong performance across datasets, suggesting that dataset size is not a limiting factor for the proposed framework.
> - **We appreciate the suggestion and plan to expand LogicTale in future releases**, incorporating additional story domains and higher-entity scenarios. This extension is straightforward within our current annotation pipeline.
>
>
>
> **Q1: Computational Cost and Refinement Steps**
>
> - **The Task is Intrinsically Hard even for Humans.**
>
>   Generating a coherent multi-panel story requires iterative refinement, causal planning, and visual consistency checking.
>   Human artists also revise panels multiple times to ensure narrative logic and visual coherence.
>   Therefore, **a certain level of iteration is inherent and expected** for this task, and is not unique to our method.
>
> - **Refinement Rates Show That Latency Is Manageable and Task-Appropriate.**
>
>   We report the proportion of refinements (edit vs. regeneration) during generation:
>
>   | Difficulty | Refinement Ratio (edit / regeneration) |
>   | ---------- | -------------------------------------- |
>   | Simple     | **0.22 / 0.04**                        |
>   | Medium     | **0.27 / 0.07**                        |
>   | Hard       | **0.38 / 0.07**                        |
>
>   These results show that:
>
>   - The overall revision cost scales reasonably with story difficulty.
>   - The latency remains well within practical limits for narrative generation.
>
>   Similar to human creative workflows, **iterative refinement is both expected and necessary**, and in our framework typically requires only **1.5–3** the time of direct generation while leading to significantly improved logical consistency.

---

> ### Author Response · Authors · 2025-11-19
> **Rebuttal (2/2)**
>
> **Q2: Robustness of LogicMiner Across Narrative Styles**
>
> - **LogicMiner is explicitly designed to extract both explicit physical causality and implicit narrative causality.**
>   Its dual-stage extraction process first identifies observable event transitions (e.g., object interactions, spatial changes), and then infers latent causal relations, such as emotional states or motivations, when they have downstream visual consequences. Therefore, LogicMiner handles cases like *“the dog, remembering his master, felt sad”* by mapping the implicit cause to visually expressible states (e.g., lowered posture, reduced activity), and it handles explicit physical causality in the same unified causal-chain representation.
> - **The subsequent ShotPlanner further strengthens robustness by providing fine-grained visual descriptions for each panel, including detailed attributes of characters’ expressions, postures, and object states.**
>   This ensures that even implicitly induced narrative states extracted by LogicMiner can be grounded into concrete visual instructions, enabling the system to express non-physical causality (e.g., emotions, intentions) through visually interpretable cues.
> - **As shown in Table 2 (ablation study), enabling LogicMiner leads to clear improvements in both Narrative Causality and Story Readability, demonstrating its effectiveness across different narrative styles.**
>   Removing LogicMiner notably degrades both causal coherence and interpretability, indicating that the agent is not narrowly tuned to explicit physical chains but is essential for handling diverse narrative forms, including those dominated by implicit or abstract causal relations.
>
>
>
> **Q3: Clarification on GPT-4-image-1 vs GPT-4o + GPT-image-1**
>
> Thank you for pointing out the naming inconsistency across Section 5.1, Table 1, and Section 5.2.1. These entries refer to the **same** baseline method.
> The baseline is a unified two-stage pipeline in which:
>
> - **GPT-4o performs story interpretation and panel planning**
> - **GPT-image-1 performs image generation**, following OpenAI’s current API structure where image generation functionality is consolidated under the GPT-image-1 endpoint.
>
> **We will update the revised PDF to consistently use the name “GPT-4o + GPT-image-1”** throughout the paper. This clarification only concerns naming; the underlying method and reported results remain unchanged.
>
>
>
> **Q4: Qualitative Failure Case**
>
> **In the updated version of the paper, we provide an example of  "Hard" level in Section5.4.** For this story, we explicitly show the **causal graph constructed by our method**, which includes multiple branches and cross-linked dependencies.
>
> - **Causal Graph Analysis:** The resulting graph successfully captures the intended narrative logic and supports correct global verification during generation. Empirically, this example demonstrates the framework’s ability to handle complex, ambiguous, and multi-threaded story developments effectively.
> - **Image Sequence Analysis:** Our framework demonstrates **better narrative alignment** compared with representative baselines such as StoryDiffusion and *GPT-4o + GPT-image-1*. But in *panel 2* and *panel 5*, the student's clothes changes, which leads to misunderstanding of the storyline.
> - **More challenging problems remain**: For example, **visual emotional expressiveness can be insufficiently subtle**, **event-level coherence may occasionally degrade**, and **maintaining consistency in multi-instance scenarios remains difficult**. These limitations stem largely from the inherent challenge of visually conveying fine-grained emotional trajectories, particularly when the narrative relies on subtle affective cues. **We believe this highlights a promising direction for future research, toward modeling emotion-aware visual logic and enhanced reasoning over nuanced narrative dynamics.**

---

### Official Review · Reviewer_GhPS · 2025-10-31

**Soundness:** 3
**Presentation:** 3
**Contribution:** 3
**Rating:** 6
**Confidence:** 3

**Summary:**

This paper presents LogiStory, a framework for generating multi-image visual stories with a focus on logical coherence. The authors introduce the concept of "visual logic" and propose a multi-agent system to plan the narrative, combined with verification modules to refine the generated images. To evaluate their approach, they construct a new benchmark, LogicTale. The paper is well-written, and the problem it addresses is both interesting and relevant to the community.

**Strengths:**

1.  **Interesting and Important Task:** The paper successfully highlights "visual logic" as a critical dimension in visual storytelling. This focus enriches the task by moving beyond per-image quality to sequence-level narrative coherence, which is an important and challenging problem.

2.  **Effective Framework Design:** The proposed framework, with its planning and verification stages, appears to be an effective solution to the problem identified. The qualitative results demonstrate its capability to produce more logically consistent visual narratives compared to several strong baselines.

3.  **Contributions of a New Benchmark:** The introduction of the LogicTale benchmark is a valuable contribution. It provides a dedicated resource for evaluating narrative logic, and the paper's experiments on this benchmark effectively showcase the performance of the proposed method.

**Weaknesses:**

1.  **Limited Novelty in Core Methodology:** While the overall system is well-engineered, the primary contribution appears to be the careful orchestration of existing, powerful models (LLMs, MLLMs, and diffusion models like Flux) into a sophisticated pipeline. The novelty in terms of fundamental model architecture or learning paradigms seems limited.

2.  **System Complexity and Robustness Concerns:** The framework's effectiveness seems heavily dependent on the consistent and reliable performance of each component model (e.g., the agents). As a multi-stage pipeline, it is susceptible to error propagation, where an issue in an early stage could negatively impact the entire sequence. Additionally, this cascaded design naturally incurs significant computational overhead and latency, which could be a practical concern.

3.  **Fairness of Experimental Comparison:** The experiments compare LogiStory, a highly customized pipeline for this specific task, against end-to-end models like GPT-4o and Gemini. This comparison, while informative, may not be entirely fair, as the proposed method leverages the combined strengths of multiple specialized models. It would be beneficial to discuss or explore fairer comparison settings, perhaps by comparing against other modular pipelines or by breaking down the performance gains attributable to each component.

**Questions:**

My main questions are related to the weaknesses detailed above. In addition, I would like to ask the following:

1. **Robustness of the MLLM-based evaluation metrics:**  For instance, during the VQA-based `Narrative Causality` assessment, how can we distinguish between failures of the "evaluator" MLLM and genuine logical flaws in the generated images? Have you conducted any analysis on the reliability of the evaluator models themselves?

2.  The Global Causal Verifier relies on a pre-constructed causal graph. How does the framework handle stories with very complex, non-linear, or ambiguous causal structures?

I would be open to raising my score should the authors address the concerns noted above.

---

> ### Author Response · Authors · 2025-11-19
> **Rebuttal (1/3)**
>
> We sincerely thank the reviewer for the insightful feedback. Below we respond to each concern raised:
>
> **W1. Novelty**
>
> We emphasize that research contributions go beyond algorithms to include **framework-level innovations** enabling new capabilities. Our **multi-agent design structures story generation via causal reasoning**, which monolithic methods lack.
>
> Similar advances appear in works like **HuggingGPT** and **Stanford Town**, where novel coordination unlocks new functions. Likewise, our framework enables **explicit narrative logic modeling across image sequences**, addressing a gap in prior focus on character consistency or image-text alignment.
>
> In particular:
>
> - **First Logic-Aware Story Generation Framework**: LogiStory is the **first** framework to explicitly model **global narrative understanding and causal logic** across image sequences. Prior methods mainly focus on **character consistency** or **scene-level image-text alignment**. In contrast, we introduce a **multi-agent planning architecture** and **logic enhancement modules** that **explicitly extract, model, and verify** both local and global causal chains, opening a new direction for **interpretable and coherent visual storytelling**.
> - **First Causally-Annotated Story Dataset**: We present **LogicTale**, the **first dataset** with **explicit causal annotations** for multi-image stories. Unlike prior datasets (e.g., *StorySalon*, *PororoSV*) that offer only raw narratives or unstructured image-text pairs, LogicTale enables **structured, logic-aware evaluation**. It also features **diverse story sources** (classic and human-authored) and **difficulty-based organization** for more granular analysis.
> - **First Evaluation Metrics for Story-Level Logic**: We propose novel automatic metrics, *Narrative Causality*, *Instance Consistency*, and *Story Readability*, designed specifically for **multi-image story logic**. These go beyond traditional metrics like **FID, CLIPScore, and ViStoryBench**, which focus on **single-image quality** and overlook sequence-level coherence. Our metrics offer a more meaningful way to assess **story integrity and interpretability**.
> - **Cross-reviewer agreement on novelty**: We are pleased that **multiple reviewers explicitly acknowledged** the **novelty and importance** of our contributions, especially the **focus on visual logic modeling** and the development of **new evaluation tools**.
>
> **W3: Fairness of Experimental Comparison**
>
> - **We would like to clarify that our experiments already include comparisons against multiple categories of baselines, including existing agent-based and modular pipelines specifically designed for story visualization.**
>   To the best of our knowledge, the only available modular/agent-based systems prior to our work are **MM-StoryAgent** and **Story-Adapter**.
>   We compare against both in **Table 1** of the main paper and **Table 4** in the Appendix. The results consistently show that LogiStory surpasses these approaches across nearly all logic-related and narrative-quality metrics, demonstrating the effectiveness of our logic-aware framework.
>
>   | **Method**    | **ICons.↑** | **NCausal.↑** | **SRead.↑** | **AesthQ.↑** | **SCons.↑** | **CExpr.↑** |
>   | ------------- | ----------- | ------------- | ----------- | ------------ | ----------- | ----------- |
>   | Story-Adapter | 3.65        | 2.23          | 0.5733      | 0.2952       | 0.8195      | 3.16        |
>   | MM-StoryAgent | 3.03        | 2.63          | 0.5787      | 0.2942       | 0.8023      | 2.54        |
>   | LogiStory     | **4.23**    | **4.45**      | **0.8267**  | **0.3088**   | **0.8572**  | **4.32**    |
>
>   These comparisons indicate that our performance gains are not solely due to strong foundation models, but arise from the **logic-aware multi-agent formulation and verification mechanisms**.
>
> - **Furthermore, the ablation studies in Table 2 and Table 3 clearly demonstrate that the improvements come from the proposed components rather than model scaling.**
>   Specifically:
>
>   - Removing the *multi-agent planning* module leads to large drops in story-level coherence, confirming that structured decomposition is essential for producing logically grounded narratives.
>   - Removing the *Local Causal Monitor* or *Global Causal Verifier* significantly reduces robustness and narrative causality, showing that the logic-aware checks are crucial for preventing error propagation and maintaining high-level story consistency.
>
>   Together, these results provide strong evidence that the performance of LogiStory is attributable to our framework design rather than unfair advantages in model strength or pipeline complexity.

---

> ### Author Response · Authors · 2025-11-19
> **Rebuttal (2/3)**
>
> **W2: System Complexity and Robustness**
>
> We appreciate the reviewer’s concern and address it from two perspectives: **System Robustness** and **Computational Cost**.
>
> **1. System Robustness**
>
> - **Explicit Intermediate Representations Reduce Error Propagation.**
>
>   Each stage in LogiStory communicates through **structured, machine-interpretable representations**, including entity tables, event tuples, causal graphs, and panel scripts.
>   These structured interfaces constrain information flow and **significantly reduce uncontrolled cascading errors**, in contrast to black-box end-to-end pipelines.
> - **The Local Causal Monitor and Global Causal Verifier Are Designed to Improve Stability.**
>
>   These two modules are not optional add-ons but **core stabilizing mechanisms**.
>   The Local Causal Monitor catches frame-level inconsistencies and performs targeted refinements, while the Global Causal Verifier enforces story-level causal compatibility.
>   Together, they **actively prevent early deviations from influencing later stages**, improving the robustness of long-horizon generation.
> - **Empirical Evidence Demonstrates Stable Performance Gains.**
>
>   **Table 1 (main results)** shows that LogiStory consistently outperforms all baselines across narrative-level metrics, including Narrative Causality and Story Readability.
>   Furthermore, **Tables 2 and 3 (ablation studies)** show that removing either the Local Causal Monitor or the Global Causal Verifier leads to **clear and consistent performance drops**, confirming that the framework improves stability rather than amplifying error propagation.
>
> **2. Computational Cost**
>
> - **The Task is Intrinsically Hard even for Humans**
>
>   Generating a coherent multi-panel story requires iterative refinement, causal planning, and visual consistency checking.
>   Human artists also revise panels multiple times to ensure narrative logic and visual coherence.
>   Therefore, **a certain level of iteration is inherent and expected** for this task, and is not unique to our method.
>
> - **Refinement Rates Show That Latency Is Manageable and Task-Appropriate**
>
>   We report the proportion of refinements (edit vs. regeneration) during generation:
>
>   | Difficulty | Refinement Ratio (edit / regeneration) |
>   | ---------- | -------------------------------------- |
>   | Simple     | **0.22 / 0.04**                        |
>   | Medium     | **0.27 / 0.07**                        |
>   | Hard       | **0.38 / 0.07**                        |
>
>   These results show that:
>
>   - The overall revision cost scales reasonably with story difficulty.
>   - The latency remains well within practical limits for narrative generation.
>
>   Similar to human creative workflows, **iterative refinement is both expected and necessary**, and in our framework typically requires only **1.5–3** the time of direct generation while leading to significantly improved logical consistency.
>
>
> **Q1: Robustness of MLLM-based Evaluation Metrics**
>
> - **Grounded in Carefully Annotated Narrative Causality**
>
>   In constructing LogicTale, we meticulously decomposed each story into **explicit causal chains**, converting complex plot progressions into **VQA-friendly, atomic cause–effect units**. This design ensures that the VQA-based evaluator operates on well-defined logical primitives rather than ambiguous narrative descriptions.
>
> - **Automatic NCausal Metric Reliability**
>
>
>   To further verify the robustness of the automatic evaluation protocol, we conducted **human validation on Narrative Causality** and additionally evaluated the metric using **Qwen2.5-VL-7B** and **Intern2.5VL-8B**, two open-source MLLMs from different model families. Their evaluation results consistently ranked **LogiStory** as the top-performing method, and their scores showed strong agreement with human ratings. These results demonstrate the **evaluation robustness across different model backbones** and further support the reliability of our automatic assessment.
>
>   | Evaluator MLLM | Correlation w/ Human NCausal.        | LogiStory Ranking |
>   | -------------- | ------------------------------------ | ----------------- |
>   | Human          | -                                    | Best              |
>   | GPT-4o         | High (Spearman=1.000, Pearson=0.975) | Best              |
>   | Qwen2.5-VL-7B  | High (Spearman=0.933, Pearson=0.977) | Best              |
>   | Intern2.5VL-8B | High (Spearman=0.900, Pearson=0.896) | Best              |
>
> - **Challenges on Fine-grained Narrative Reasoning**
>
>   For tasks requiring **fine-grained causal and narrative understanding across modalities** (e.g., story visualization, text-based narratives, video temporal reasoning), evaluation metrics that are both reliable and model-agnostic are still underexplored. We view the construction of such metrics as an **important and promising research direction**, and our work provides a concrete step toward this by operationalizing story-level causal chains for visual narratives.

---

> ### Author Response · Authors · 2025-11-19
> **Rebuttal (3/3)**
>
> **Q2: Handling Complex or Non-linear Causal Structures**
>
> - **The Global Causal Verifier is not restricted to linear causal chains**
>   It constructs a **directed multi-graph** rather than a simple sequence, which enables it to capture:
>
>   - **branching causal structures** where a single event leads to multiple diverging outcomes;
>   - **converging causal structures** where multiple preconditions jointly trigger a subsequent event;
>   - **parallel or interleaved events** that evolve simultaneously without strict temporal ordering.
>
>   Concretely, each event is represented as a tuple of {actor, action, target, result}, and edges are added whenever the postconditions of one event satisfy the preconditions of another. This representation allows the system to recover complex developmental relationships in narratives, including conditional dependencies and multi-path progressions.
>
> - **A Complex Example with Non-linear and Interdependent Causal Dynamics**
>
>   **In the updated version of the paper, we provide an example in Section5.4.** For this story, we explicitly show the **causal graph constructed by our method**, which includes multiple branches and cross-linked dependencies.
>
>   - **Causal Graph Analysis:** The resulting graph successfully captures the intended narrative logic and supports correct global verification during generation. Empirically, this example demonstrates the framework’s ability to handle complex, ambiguous, and multi-threaded story developments effectively.
>
>   - **Image Sequence Analysis:** Our framework demonstrates **better narrative alignment** compared with representative baselines such as StoryDiffusion and *GPT-4o + GPT-image-1*. However, certain challenges remain. For instance, **visual emotional expression may be insufficiently delicate**, and **event-level coherence can occasionally degrade**. This is partly due to the inherent difficulty of visually presenting fine-grained emotional trajectories, especially when narrative intent relies strongly on subtle affective cues.
>
>   - **Potential Improvement**
>
>     - **Modular causal graph:** Split the story into multiple subplot-level causal subgraphs (e.g., per character or storyline), with synchronization edges to manage interactions across subplots.
>
>     - **Temporal encoding & non-linear support:** Assign temporal labels and relative ordering to each event/state to support flashbacks or non-linear narration, potentially via versioned nodes or timeline constructs.
>
>     - **Character-centric reasoning:** Maintain per-character state trackers and use GNN-based propagation to detect and resolve cross-character inconsistencies.
>
>     - **Scalable planning:** Use chunked planning with lazy subplot expansion (activating branches only when needed) and attention-based routing between agents to keep computation efficient.

---

### Official Review · Reviewer_oUdB · 2025-11-01

**Soundness:** 3
**Presentation:** 3
**Contribution:** 3
**Rating:** 6
**Confidence:** 4

**Summary:**

This paper introduces the concept of "visual logic" to address the challenge of generating causally and perceptually coherent visual story sequences. The authors define visual logic as the perceptual and causal coherence among characters, actions, and scenes over time. To tackle this, they propose `LogiStory`, a framework composed of two main parts:
1.  **Logic-Aware Multi-agent System:** This system performs structured narrative planning. It uses three agents (`SceneCrafter`, `LogicMiner`, `ShotPlanner`) to decompose a story text $S$ into character/object definitions $E$, extract a causal chain of key events $K$, and plan a sequence of detailed visual shot specifications $P = \\{p_1, ..., p_T\\}$.
2.  **Visual Logic Enhancement Module:** This module enforces coherence during the generation of the image sequence $I = \\{I_1, ..., I_T\\}$. It includes:
    *   A **Local Causal Monitor** that checks step-by-step narrative plausibility by calculating a causal score $\\psi_t = C_p(I_t | M_{t-1})$ based on a memory buffer $M_{t-1}$.
    *   A **Global Causal Verifier** that uses a story-level causal graph $G_{causal}$ to validate state transitions and guide an image refinement process based on predefined thresholds $\\tau_1$ and $\\tau_2$.

To facilitate development and evaluation, the authors construct `LogicTale`, a new benchmark of 60 stories with rich annotations for causal events. They also propose a comprehensive evaluation protocol that measures "visual logic" and perceptual quality. Experiments show that `LogiStory` significantly outperforms strong open-source and closed-source baselines, particularly in metrics related to narrative logic and readability.

**Strengths:**

1.  **Novel and Important Problem Formulation:** The paper's primary strength is its formalization and focus on "visual logic." This provides a clear, principled target for a key failure mode of current generative models—the lack of narrative and causal coherence in sequences.
2.  **Well-Designed and Interpretable Framework:** `LogiStory` is a thoughtfully engineered system. The multi-agent planner provides an interpretable intermediate representation, and the dual-level verification (local and global) robustly enforces logical consistency.
3.  **Comprehensive and Rigorous Evaluation:** The experimental evaluation is a major strength. The authors compare against a wide range of powerful baselines, including top-tier commercial models, and use a combination of automatic and human studies for robust validation.
4.  **Creation of a Valuable Benchmark:** `LogicTale` is a significant contribution in its own right, enabling the community to move beyond simple frame-level metrics and systematically evaluate the storytelling capabilities of generative models.
5.  **Exceptional Clarity and Presentation:** The paper is written with outstanding clarity. Complex ideas are explained intuitively, and the high-quality figures and detailed appendix make the work easy to understand and reproduce.

**Weaknesses:**

1.  **Scale of the LogicTale Benchmark:** The most apparent weakness is the scale of the `LogicTale` benchmark (60 stories). While the authors justify this with a saturation analysis, a larger dataset would provide stronger evidence for the generalizability of the findings.
2.  **Reliance on MLLMs for Automatic Evaluation:** The automatic evaluation protocol for core visual logic metrics relies on judgments from MLLMs, which raises concerns about potential evaluation bias (an "LLM-evaluating-LLM" loop). While mitigated by high correlation with human ratings, this is a methodological limitation.
3.  **Complexity and Latency:** `LogiStory` is a multi-stage pipeline involving several LLM/MLLM calls, which likely entails significant computational cost and latency. A discussion on the framework's efficiency and robustness to component failures would be beneficial.
4.  **Insufficient Ablation of Refinement Mechanism:** The paper states that refinement is done via "image editing tools" for scores in the range $[\\tau_1, \\tau_2)$. A more detailed analysis or ablation of this step (e.g., frequency of use, comparison to simple regeneration) would strengthen the paper.

**Questions:**

1.  **On Evaluation and Potential Bias:** Did you experiment with different families of MLLMs for evaluation (e.g., using an open-source model to evaluate proprietary model outputs) to test the robustness of the evaluation scores?
2.  **On the `LogicMiner`'s Robustness:** The `LogicMiner` agent is crucial for constructing the causal graph. How robust is its capability to infer implicit events, especially for subtle logic? What are its common failure modes, and how does the framework handle an incorrect causal graph?
3.  **On Verifier Interaction:** In Table 3, the `Instance Consistency` score for "Ours (w/ local)" is 4.24, while for "Ours (w/ both)" it is 4.23. This is a minor, perhaps insignificant, difference. Could you offer any insight into why adding the `Global Causal Verifier` might not strictly improve this specific metric?
4.  **On the Refinement Process:** Regarding the image refinement step triggered when $\\psi_t \\in [\\tau_1, \\tau_2)$, could you provide statistics from your experiments on the frequency of this action (accept vs. refine vs. regenerate)? What is the computational overhead of this step compared to a simple regeneration?
5.  **On Scalability:** How do you envision the framework scaling to more complex narratives with many characters, parallel plotlines, or non-linear temporal structures? Would the current causal graph representation be sufficient?

---

> ### Author Response · Authors · 2025-11-19
> **Rebuttal (1/3)**
>
> We sincerely thank the reviewer for the insightful feedback. Below we respond to each concern raised:
>
> **W1: Dataset Scale**
>
> We thank the reviewer for raising concerns about dataset size and generalizability. Below we clarify our design rationale from three perspectives:
>
> - **Benchmark Size Comparison with Prior Works**
>
>   Regarding **evaluation scale**, our dataset is **comparable or larger** than prior works: *StoryDiffusion* (20 short single-character stories), *ConsiStory* (20 handcrafted scenes), *MM-StoryAgent* (100 LLM-generated stories), *MovieAgent* (12 authored examples), and *ViStoryBench* (80 test cases). Thus, **LogicTale’s 60 richly annotated, multi-character stories with varied difficulty** provide a **solid, representative benchmark**.
>
> - **Empirical Study on Dataset Size (Appendix §C.3)**:
>
>   We conducted an ablation study by incrementally increasing the dataset size from 12 to 60. Results show that **metric stability (Kendall’s Tau) is reached around 36–40 stories**, across six evaluation dimensions. This demonstrates that **reliable evaluation and ranking are achievable** with our current dataset size.
>
> - **Diversity & Quality in Composition (Appendix §C.1)**:
>   LogicTale balances **diversity** and **annotation quality** by including both **classic** (e.g., Aesop’s Fables, folktales) and **human-authored** stories, ensuring cultural variety and reducing evaluation bias. All stories are grouped by **difficulty levels** for fine-grained analysis. Despite its smaller size, LogicTale is the **first dataset** with **structured causal annotations** for multi-image story visualization, enabling logic-aware generation and evaluation, which existing datasets cannot support.
>
> We agree that future scaling of LogicTale would benefit the community, and we are actively working on releasing a larger version with more diverse domains.
>
>
>
> **W2&Q1: Discussion on MLLM-based Evaluation Metrics**
>
> - **Grounded in Carefully Annotated Narrative Causality**
>   In constructing LogicTale, we meticulously decomposed each story into **explicit causal chains**, converting complex plot progressions into **VQA-friendly, atomic cause–effect units**. This design ensures that the VQA-based evaluator operates on well-defined logical primitives rather than ambiguous narrative descriptions.
>
> - **Automatic NCausal Metric Reliability**
>
>   To further verify the robustness of the automatic evaluation protocol, we conducted **human validation on Narrative Causality** and additionally evaluated the metric using **Qwen2.5-VL-7B** and **Intern2.5VL-8B**, two open-source MLLMs from different model families. Their evaluation results consistently ranked **LogiStory** as the top-performing method, and their scores showed strong agreement with human ratings. These results demonstrate the **evaluation robustness across different model backbones** and further support the reliability of our automatic assessment.
>
>   | Evaluator MLLM | Correlation w/ Human NCausal.        | LogiStory Ranking |
>   | -------------- | ------------------------------------ | ----------------- |
>   | Human          | -                                    | Best              |
>   | GPT-4o         | High (Spearman=1.000, Pearson=0.975) | Best              |
>   | Qwen2.5-VL-7B  | High (Spearman=0.933, Pearson=0.977) | Best              |
>   | Intern2.5VL-8B | High (Spearman=0.900, Pearson=0.896) | Best              |
>
> - **Challenges on Fine-grained Narrative Reasoning**
>   For tasks requiring **fine-grained causal and narrative understanding across modalities** (e.g., story visualization, text-based narratives, video temporal reasoning), evaluation metrics that are both reliable and model-agnostic are still underexplored. We view the construction of such metrics as an **important and promising research direction**, and our work provides a concrete step toward this by operationalizing story-level causal chains for visual narratives.

---

> ### Author Response · Authors · 2025-11-19
> **Rebuttal (2/3)**
>
> **W3&W4&Q4: Computational Cost&Refinement Process Statistics**
>
> - **The Task is Intrinsically Hard even for Humans**
>   Generating a coherent multi-panel story requires iterative refinement, causal planning, and visual consistency checking.
>   Human artists also revise panels multiple times to ensure narrative logic and visual coherence.
>   Therefore, **a certain level of iteration is inherent and expected** for this task, and is not unique to our method.
>
> - **Refinement Rates Show That Latency Is Manageable and Task-Appropriate.**
>   We report the proportion of refinements (edit vs. regeneration) during generation:
>
>   | Difficulty | Refinement Ratio (edit / regeneration) |
>   | ---------- | -------------------------------------- |
>   | Simple     | **0.22 / 0.04**                        |
>   | Medium     | **0.27 / 0.07**                        |
>   | Hard       | **0.38 / 0.07**                        |
>
>   These results show that:
>
>   - The overall revision cost scales reasonably with story difficulty.
>   - The latency remains well within practical limits for narrative generation.
>
>   Similar to human creative workflows, **iterative refinement is both expected and necessary**, and in our framework typically requires only **1.5–3** the time of direct generation while leading to significantly improved logical consistency.
>
>
>
>
>
> **Q2: Regarding the Robustness of `LogicMiner`**
>
> - **`LogicMiner` operates on finely segmented narrative fragments**
>
>   `LogicMiner` performs local reasoning on compact **scene units avoids long-range ambiguity**. It improves stability when inferring implicit and subtle causal events, as reasoning is constrained within structurally coherent story segments.
>
> - **Causal refinement mechanism and multi-agent validation loop.**
>
>   - Both Local Causal Monitor and Global Causal Verifier cross-check extracted causal relations before final rendering. The empirical refinement ratio indicates that many inconsistencies are identified and resolved through iterative correction.
>   - **Table 3 shows the causal-graph-guided refinement significantly improves narrative causality and story readability**, validating the system’s ability to recover from minor extraction errors.
>
> - **Common failure modes mainly arise from causal granularity inconsistency.**
>
>   - Sometimes a complex transformation is **over-fragmented into multiple micro-events**. In other cases, **multiple implicit steps are merged into a single simplified transition**.
>   - This challenge intensifies with **higher story complexity**, as causal decomposition is inherently difficult. **Table 5 further supports this observation**, showing that **stronger backbone models lead to improved causal extraction quality**, suggesting that enhanced reasoning capabilities help mitigate such errors.
>   - We recognize this as **a valuable future direction**, where multi-level causal abstraction or trainable causal segmentation models could enhance robustness.
>
>
>
>
>
>
>
> **Q3: Instance Consistency and Verifier Interaction**
>
> We appreciate the reviewer’s observation regarding the slight difference in *Instance Consistency* between **“Ours (w/ local)” (4.24)** and **“Ours (w/ both)” (4.23)**. We provide our clarification below:
>
> - **The Global Causal Verifier is primarily designed to enforce causal consistency and state transition correctness rather than low-level appearance or pixel alignment.**
>   In a few cases, when a causal conflict is detected, the verifier triggers targeted edits (e.g., re-framing or relocation of entities), which may unintentionally introduce minor visual deviations, hence slightly affecting the *Instance Consistency* metric.
> - **The observed difference (4.24 vs. 4.23) is small and likely falls within the natural evaluation variance.**
>   We verified that this deviation is within the noise range of both automated and human assessments. In contrast, adding the Global Verifier leads to clear improvements in **Narrative Causality** and **Story Readability**, which are the primary focus of our framework.
>
> Overall, **the Global Causal Verifier provides substantial benefits to narrative coherence**, and the marginal fluctuation in *Instance Consistency* does not indicate performance degradation but rather reflects a causal reasoning trade-off that is well compensated at the story level.

---

> ### Author Response · Authors · 2025-11-19
> **Rebuttal (3/3)**
>
> **Q5: Handling Complex or Non-linear Causal Structures**
>
> - **The Global Causal Verifier is not restricted to linear causal chains.**
>   It constructs a **directed multi-graph** rather than a simple sequence, which enables it to capture:
>
>   - **branching causal structures** where a single event leads to multiple diverging outcomes;
>   - **converging causal structures** where multiple preconditions jointly trigger a subsequent event;
>   - **parallel or interleaved events** that evolve simultaneously without strict temporal ordering.
>
>   Concretely, each event is represented as a tuple of {actor, action, target, result}, and edges are added whenever the postconditions of one event satisfy the preconditions of another. This representation allows the system to recover complex developmental relationships in narratives, including conditional dependencies and multi-path progressions.
>
> - **A Complex Example with Non-linear and Interdependent Causal Dynamics**
>
>   **In the updated version of the paper, we provide an example in Section 5.4.** For this story, we explicitly show the **causal graph constructed by our method**, which includes multiple branches and cross-linked dependencies.
>
>   - **Causal Graph Analysis:** The resulting graph successfully captures the intended narrative logic and supports correct global verification during generation. Empirically, this example demonstrates the framework’s ability to handle complex, ambiguous, and multi-threaded story developments effectively.
>
>   - **Image Sequence Analysis:** Our framework demonstrates **better narrative alignment** compared with representative baselines such as StoryDiffusion and *GPT-4o + GPT-image-1*. However, certain challenges remain. For instance, **visual emotional expression may be insufficiently delicate**, and **event-level coherence can occasionally degrade**. This is partly due to the inherent difficulty of visually presenting fine-grained emotional trajectories, especially when narrative intent relies strongly on subtle affective cues.
>
>   - **Potential Improvement**
>
>     - **Modular causal graph:** Split the story into multiple subplot-level causal subgraphs (e.g., per character or storyline), with synchronization edges to manage interactions across subplots.
>
>     - **Temporal encoding & non-linear support:** Assign temporal labels and relative ordering to each event/state to support flashbacks or non-linear narration, potentially via versioned nodes or timeline constructs.
>
>     - **Character-centric reasoning:** Maintain per-character state trackers and use GNN-based propagation to detect and resolve cross-character inconsistencies.
>
>     - **Scalable planning:** Use chunked planning with lazy subplot expansion (activating branches only when needed) and attention-based routing between agents to keep computation efficient.

---

> > ### Comment · Reviewer_oUdB · 2025-11-27
> > **Response to Rebuttal**
> >
> > I have read the authors' rebuttal and the other reviews. The authors have satisfactorily addressed my concerns. I believe this paper makes a solid contribution to the field, and I will maintain my initial score to support its acceptance.

---

> > > ### Author Response · Authors · 2025-11-27
> > >
> > > We sincerely thank the reviewer for their positive feedback and for recognizing the value of our contribution. We are grateful that our rebuttal has satisfactorily addressed your concerns. Thank you again for your support.

---

### Comment · Area_Chair_3nKG · 2025-11-26
**Please take a moment to read the authors’ responses**

Dear Reviewers,

I hope this message finds you well. As the Area Chair for this submission, I would like to kindly remind you that the author rebuttal is now available. Please take a moment to read the authors’ responses and, if necessary, update your reviews accordingly.

Thank you very much for your time and for contributing to the quality of ICLR 2026.

Best regards,

Your AC

---

### Comment · Area_Chair_3nKG · 2025-11-28

Dear authors and reviewers,

Please remain professional and refrain from being influenced by the event. If anyone violates the rules, please let me know, and I will flag and report it to the Program Chairs.

Your AC.

---

### Author Response · Authors · 2025-12-02
**Summary of recognized strengths and rebuttal**

## Recognized Strengths and Contributions

We thank the AC and reviewers for their careful reading. Reviewers consistently recognized that our submission makes meaningful contributions to multi-image story visualization:

1. **Novel and well-motivated agentic framework.**

   LogiStory introduces a modular multi-agent decomposition (SceneCrafter, LogicMiner, ShotPlanner) that mirrors human story production and provides structured intermediates for controlled generation. In addition, the Visual Logic Enhancement modules bring explicit logical modeling and monitoring into the pipeline, enabling narrative-level causal reasoning beyond standard generation frameworks.
2. **Explicit modeling of visual logic.**

   Our Visual Logic Enhancement module (Local Causal Monitor + Global Causal Verifier)   narrative-level causal reasoning, turning story coherence from an implicit byproduct into an explicit objective.
3. **Dataset and metrics that fill an evaluation gap.**

   LogicTale is the first multi-image story dataset with structured causal annotations and weighted event tuples, together with our Narrative Causality / Instance Consistency / Story Readability metrics, it enables fine-grained, logic-aware evaluation beyond standard single-image metrics.
4. **Strong empirical generation results.**

   Across automatic metrics, ablations, and a human study, LogiStory consistently improves narrative causality and story readability compared to both modular and end-to-end baselines.

------

## Summary of Rebuttal

Below we summarize how we addressed key reviewer concerns in the rebuttal and in the revised PDF.

1. **Efficiency & complexity.**

   We acknowledge the added computation, but given the inherent difficulty of multi-image story generation, this represents a **reasonable and competitive trade-off**. Our experiments show that **structured intermediates effectively prevent error propagation** and **enable logic-guided corrections**, while keeping latency **manageable**: the typical end-to-end runtime is only **1.5–3** the time of a single-generation call yet yields **substantially stronger narrative coherence**.
2. **Robustness of the system.**

   Our ablations (**Table 3**) and cross-backbone results (**Table 3 and Table 5**) show that each module contributes to consistent gains in narrative metrics, and LogiStory remains strong even with smaller backbones. Robustness comes from **explicit intermediate representations** that reduce error propagation and from the **Local Monitor & Global Verifier** loop, which actively detects and corrects extraction or generation errors.
3. **Reliability of MLLM-based metrics.**

   To mitigate “LLM-evaluating-LLM” concerns, we validated automatic NCausal scores against human judgments (**Spearman=1.000, Pearson=0.975**) and reproduced evaluations with multiple MLLMs (including open-source **Qwen2.5-VL-7B** and **Intern2.5VL-8B**). Different evaluators consistently rank LogiStory highest and show strong agreement with human ratings. This strong alignment is enabled by the **fine-grained causal decomposition and detailed event-level annotations** in LogicTale, which make **NCausal evaluation both precise and interpretable**.
4. **Handling complex / non-linear narratives.**

   We clarified that the **Global Causal Verifier uses a directed multi-graph** (not a simple chain) to capture **branching, converging, and parallel event structures**. We added an **illustrative hard-case example** (Section 5.4) showing multi-branch causal graphs and discussed remaining limitations (e.g., subtle emotional expression, multi-instance identity slips), and we proposed **concrete extensions for future work**.

---

### Meta-Review · Area_Chair_rgUf · 2026-01-10

**Summary:**

The recommendation leans toward acceptance given that the overall signal is positive: one reviewer is clearly supportive (wuEE: 8), and two reviewers are above-threshold but cautious (oUdB: 6, GhPS: 6). The main remaining concern is the weaker score from dd8Q: 4, which highlights risks around practicality and generalization.

The key concerns that informed the decision discussion are:

System complexity / robustness / cost — raised most strongly by dd8Q (4) and also by GhPS (6).

Dataset scale / generalizability — raised by oUdB (6) and dd8Q (4) (LogicTale size).

Evaluation reliability and fairness of comparisons — raised by oUdB (6) and GhPS (6) (MLLM-based metrics, and comparison of a tailored pipeline vs end-to-end models).

Importantly, the rebuttal addressed a substantial portion of the actionable concerns (notably acknowledged by oUdB, who explicitly maintained their supportive score post-rebuttal), leaving remaining issues largely as trade-offs rather than fatal flaws.

**Reviewer Concerns:**

oUdB: Most concerns were addressed in rebuttal: dataset scale rationale + saturation study, metric reliability via multi-MLLM + human correlation, and added refinement/latency stats + robustness discussion. Engaged after rebuttal: Yes — explicitly said concerns were satisfactorily addressed and kept score to support acceptance.

GhPS: Rebuttal partially addressed fairness/complexity/metric-robustness concerns (added modular baselines, ablations, and some discussion), but the “novelty is mostly orchestration” concern likely remains somewhat outstanding (framing helps, but may not change their core view). Engaged after rebuttal: Not shown (no post-rebuttal comment in the thread you pasted).

dd8Q: Rebuttal addressed the main concrete asks: provided refinement/edit vs regen rates and clarified the baseline naming; also added a hard-case / failure case and discussed LogicMiner robustness. Remaining: they may still view dataset scale / pipeline complexity as a concern, but it’s at least answered. Engaged after rebuttal: Not shown.

wuEE: Rebuttal addressed coupling-to-metrics and prior work (VinaBench) by clarifying claims and emphasizing external eval + human study; also confirmed release plan upon acceptance. Remaining: pipeline complexity is acknowledged but not fully “solved” (more of a trade-off). Engaged after rebuttal: Not shown.

**Reviewer Scores:**

Reviews are generally of good quality

---

### Decision · Program_Chairs · 2026-01-26

Accept (Poster)